# TVDRNet: Text-driven Viewpoint Optimization via Differentiable Rendering for 3D Reasoning Segmentation

**Zaiyang Yu** [1]  **Changshuo Wang** [2]  **Pinjie Xu** [3]  **Zhang Huang** [4]  **Yuan Shi** [1 5]  **Linjun Sun** [1 5]  **Weijun Li** [1 5]

## Abstract

Three-dimensional (3D) reasoning segmentation aims at segmenting target objects based on text instructions and 3D spatial cues. Recent efforts in 3D reasoning leverage Multimodal Large Language Models (MLLMs) to bridge the gap between text and 3D data. However, since MLLMs are primarily trained on text-image pairs, directly adapting them to unstructured 3D point clouds often fails to capture implicit semantic intent and reliably localize objects. This paper introduces TVDRNet to address these challenges. Inspired by Active Vision theory, where humans selectively choose optimal viewpoints to better observe targets, TVDRNet employs a differentiable renderer to simulate this active process in 3D perception. By using text instructions as supervision to optimize intrinsic and extrinsic rendering parameters, the TVDRNet identifies the optimal viewpoints for observing the 3D scene, and therefore learning 'where to look' based on what the text instruction 'asked to find'. This process generates informative, task-relevant 2D images that are compatible with MLLMs. TVDRNet comprises: (1) the AVPL module, establishing a learnable mapping from semantics to optimal rendering parameters; and (2) the MGL module, fusing multi-modalities via semantic grouping to guide mask generation. Experiments show TVDRNet achieves the state-of-the-art performance in 3D reasoning segmentation (Reason3D, Instruct3D) and 3D visual grounding (ScanRefer) benchmarks.

---

[1]AnnLab, Institute of Semiconductors, Chinese Academy of Sciences, Beijing, China. [2]Department of Computer Science, University College London, London, United Kingdom. [3]China University of Mining Technology - Beijing, Beijing, China. [4]School of Computer Science, Beihang University, Beijing, China. [5]University of Chinese Academy of Sciences, Beijing, China. Correspondence to: Weijun Li <wjli@semi.ac.cn>.

*Proceedings of the 43rd International Conference on Machine Learning*, Seoul, South Korea. PMLR 306, 2026. Copyright 2026 by the author(s).

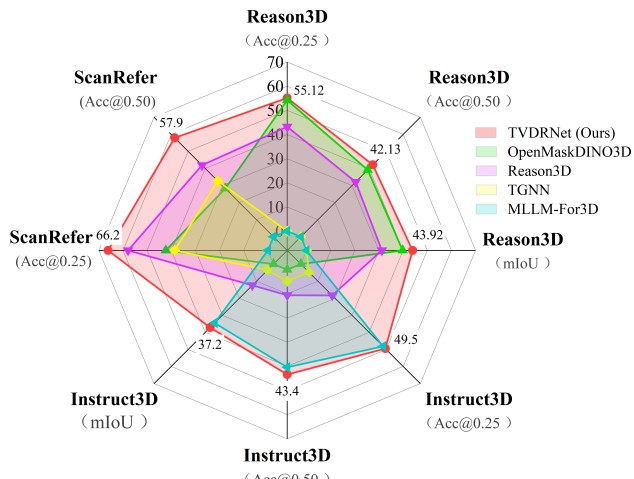

*Figure 1.* The performance of TVDRNet in 3D reasoning segmentation benchmarks (Reason3D(Huang et al., 2025b), Instruct3D(He et al., 2024)), and 3D Visual grounding (ScanRefer (Chen et al., 2020)) benchmarks.

## 1. Introduction

Human perception of the three-dimensional (3D) world stems from the brain's construction of a 3D representation from the two-dimensional (2D) projections received by retina (Yilmaz et al., 2025). When given a specific task, such as "find the car keys on the messy desk", humans do not scan the whole 3D scene; instead, the task instruction guides an active vision process in which the task-specific 2D projections are chosen to facilitate subsequent decision-making for executing the task (Moores et al., 2003), detailed in Fig. 2(a). Perhaps Artificial Intelligence (AI) systems could potentially obtain similar capabilities, enhancing the understanding of 3D scenes by selectively choosing task-relevant 2D viewpoints for observation.

This work focuses on 3D reasoning segmentation. Drawing a parallel to the "find keys" example, the specific task here is to enhance the capabilities of AI systems to "find" target objects within complex 3D scenes by analyzing the deep implications embedded in language instructions (Huang et al., 2025b). However, interpreting language instructions for 3D scenes presents significant challenges: (1) human

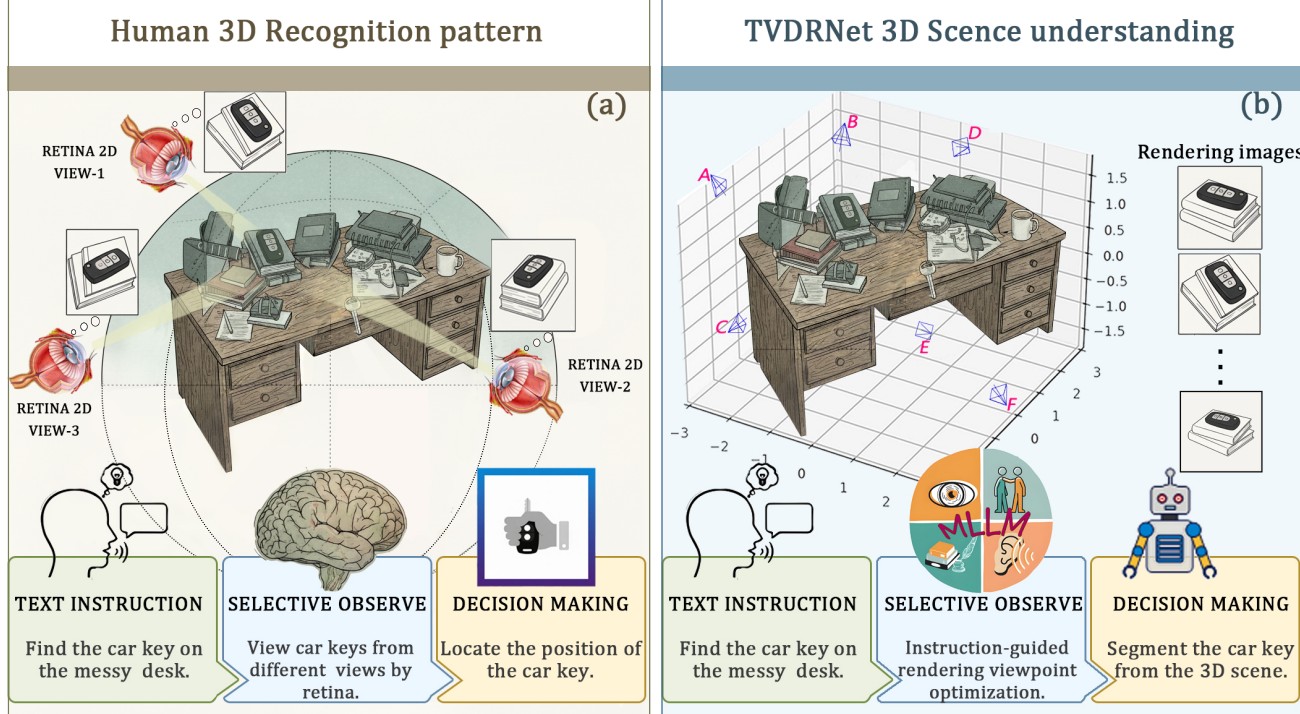

*Figure 2.* Active Vision for 3D Scene Understanding. (a) The human active vision process. Guided by textual instructions (e.g., "find the car key"), humans selectively adjust their viewpoints to observe a target object from multiple perspectives. This information-gathering step facilitates subsequent decision-making, such as locating the object. (b) The proposed TVDRNet framework. TVDRNet interprets the text instruction using a Multimodal Large Language Model (MLLM). The MLLM guides differentiable renderers (A-F) to optimize viewpoints and render task-relevant images. By integrating these rendered multi-views with the original 3D point cloud data, TVDRNet performs the required tasks (e.g., 3D segmentation, 3D Visual Grounding).

instructions are often ambiguous and open to multiple interpretations (Dan et al., 2025); (2) the sparsity and irregular geometry of point-cloud data complicate precise segmentation (Huang et al., 2025a). Addressing these challenges requires a 3D learning paradigm that effectively integrates high-level semantic language with low-level 3D geometric structures.

The emergent abilities of Multimodal Large Language Models (MLLMs, AI systems capable of processing and understanding multiple data modalities, including text, images, and 3D data) advance AI systems for 3D scene understanding. By bridging the gap between language and vision, MLLMs can partially overcome the ambiguity of human instructions and the unstructured nature of 3D point clouds (Jiang et al., 2024). However, extending MLLMs' capabilities to 3D reasoning remains a significant challenge. The challenges are twofold: (1) Erroneous localization: Due to the severe scarcity of 3D scene-language training data, MLLMs frequently generate text responses that are semantically unrelated to the 3D scene content (Zhu et al., 2025b). (2) Boundary ambiguity: The inherent sparsity and unstructured nature of 3D point clouds mean that directly adapting MLLMs for 3D reasoning segmentation often yields sub-

optimal results, such as coarse object boundary delineation (Zhang, 2025). Thereby, MLLMs partially improve the performance of 3D reasoning segmentation, but their segmentation results remain hindered by Erroneous localization and Boundary ambiguity, leading to coarse and inaccurate object segmentation.

The limitations of existing methods prompt us to draw inspiration from the pattern of human perception in 3D scenes. Inspired by how humans selectively shift their focus to gain optimal perspectives for specific tasks (mentioned in the first paragraph), we propose a 3D learning paradigm that simulates this process by computationally determining the informative 2D virtual viewpoints for rendering, in turn building a complete 3D perception that alleviates the challenge of Erroneous localization and Boundary ambiguity, detailed in Fig. 2(b).

To implement this, we introduce TVDRNet (Text-driven Viewpoint optimization with Differentiable Rendering for 3D Reasoning Segmentation). The TVDRNet employs textual instructions as supervisory signal, utilizing a differentiable renderer (a rendering system that allows gradients to flow through the rendering process, enabling end-to-end optimization of camera parameters (Hamdi et al., 2025)) to

guide the system to observe the 3D scene from optimal viewpoints dictated by the text's meaning. Unlike previous 3D reasoning segmentation methods that require the MLLMs to interpret raw data like point clouds or RGB images directly, the TVDRNet generates a semantically-aligned visual representation. This provides the MLLM with a holistic and unambiguous input that mitigates the challenges of Erroneous localization and Boundary ambiguity.

Specifically, TVDRNet integrates an Adaptive Viewpoint Position Learning (AVPL) module and a Multi-modal Group Learning (MGL) module: (1) Within the AVPL module, a differentiable renderer is guided by textual instructions to optimize rendering camera intrinsic (e.g., focal length) and extrinsic (e.g., azimuth, elevation) parameters. This process enables the iterative rendering of a 3D object from multiple viewpoints, ultimately selecting the optimal views to observe the 3D scene. The AVPL module learns a direct mapping from text to rendering parameters, preventing Erroneous localization by keeping the TVDRNet focused on the correct region of interest in the 3D scene. (2) The MGL module is designed to fully leverage the multi-view images generated by AVPL. It partitions features from the point cloud and the rendered multi-views, organizing them into semantic groups based on their inherent correlations. Through the fusion of these groups, MGL integrates the point cloud's precise spatial structure with the holistic textural details from the multi-views. This process yields a holistic 3D representation that effectively resolves boundary ambiguity, enabling TVDRNet to segment object boundaries with high precision, as shown in Fig. 1. The contributions are threefold:

1) We propose the AVPL module, which employs a text-guided differentiable renderer to optimize camera parameters. This approach establishes a direct mapping from language semantics to optimal viewpoints, mitigating erroneous localization in 3D scenes.

2) We propose the MGL module, which partitions and fuses features from point clouds and multi-view images into semantic groups. This method effectively integrates precise geometric structure (point clouds) with holistic textural details (multi-views) to overcome boundary ambiguity.

3) Experiments are conducted on 3D reasoning segmentation (Reason3D, Instruct3D) and 3D visual grounding (ScanRefer) tasks, where TVDRNet achieves state-of-the-art performance in both of 3D reasoning and visual grounding benchmarks.

**Conflict of Interest Disclosure.** The authors have no conflicts of interest to disclose.

## 2. Related work

**Multimodal Large Language Model in 3D Understanding.** Multimodal large language models (MLLMs), language models conditioned on non-text inputs, have been extended from images to 3D scenes. Early 3D-language systems studied scene grounding and instruction following through 3D-LLM (Hong et al., 2023), PointLLM (Xu et al., 2024), Chat-3D (Wang et al., 2023), LEO (Huang et al., 2024b), and LL3DA (Chen et al., 2024a). These works connect point clouds, object proposals, or scene tokens with language decoders, but they mainly address description, question answering, or grounding. Recent studies expand the setting: 3UR-LLM (Xiong et al., 2025) supports end-to-end 3D scene understanding, SpatialLM (Mao et al., 2025) targets structured indoor modeling, PointLLM-V2 (Xu et al., 2025) improves point-cloud-language alignment, and 2026 works such as Direction-aware 3D Large Multimodal Models (Liu et al., 2026a) and MAG-3D (Zheng et al., 2026) focus on ego-pose-aware spatial reasoning and grounded multi-agent reasoning. Our work follows this MLLM-based 3D understanding line, but uses it for reasoning segmentation rather than only scene-level dialogue or localization.

**Differentiable Rendering in 3D Understanding.** Differentiable rendering, rendering that supports gradient backpropagation, provides a link between 3D representations and image supervision. Neural 3D Mesh Renderer (Kato et al., 2018) and Soft Rasterizer (Liu et al., 2019) introduced gradient-based mesh rasterization for image-based 3D reasoning, while PyTorch3D (Ravi et al., 2020) made these operators practical in learning pipelines. NeRF (Mildenhall et al., 2022), IDR (Yariv et al., 2020), NeuS (Wang et al., 2021), and VolSDF (Zhang et al., 2023) further connected implicit geometry with differentiable image formation. 3D Gaussian Splatting (Kerbl et al., 2023) changed the representation toward explicit Gaussian primitives, and later work including 3DGUT (Wu et al., 2025), Volumetrically Consistent 3D Gaussian Rasterization (Talegaonkar et al., 2025), StochasticSplats (Kheradmand et al., 2025), Diff3R (Liu et al., 2026b), and MuSASplat (Xu et al., 2026) studies camera models, consistency, probabilistic splatting, reconstruction, and sparse-view rendering. Our work is connected to this line because it uses rendering as a 2D-3D bridge, but differs by using the rendered evidence to support language-conditioned 3D segmentation.

**3D Reasoning Segmentation.** 3D reasoning segmentation means selecting scene regions according to a language query that may require object attributes, relations, or context. Earlier grounding datasets and models, including ScanRefer (Chen et al., 2020), ReferIt3D (Achlioptas et al., 2020), 3DVG-Transformer (Zhao et al., 2021), and BUTD-DETR (Jain et al., 2022), established 3D object localization from text. OpenMask3D (Takmaz et al., 2023) moved

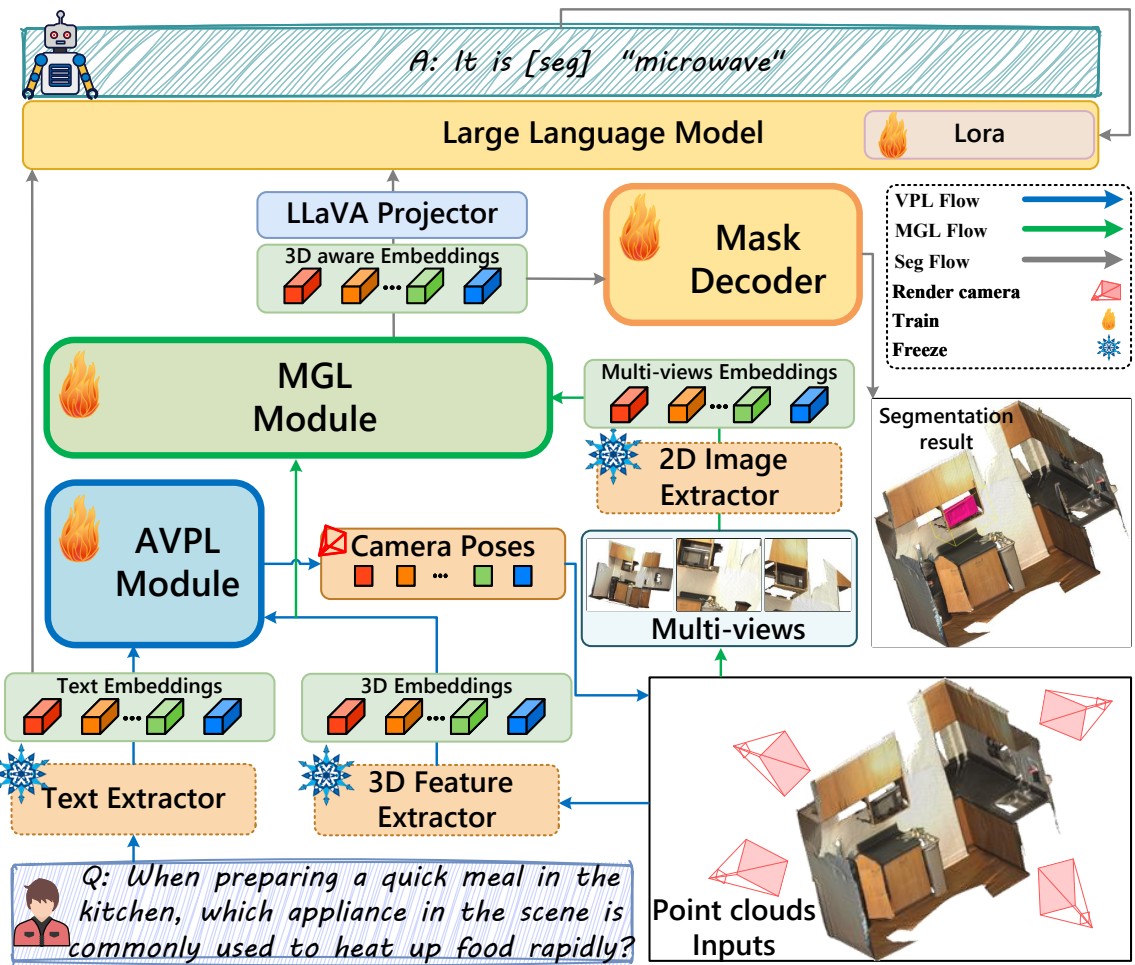

*Figure 3.* The TVDRNet Framework. The framework comprises three main components. (1) **Adaptive Viewpoint Position Learning, AVPL (blue arrows)**. The AVPL stage processes the input text and point cloud to predict optimal rendering intrinsic (e.g., focal length) and extrinsic (e.g., azimuth, elevation) parameters, guiding a differentiable renderer to generate query-specific multi-view images. (2) **Multi-modal Group Learning, MGL (green arrows)**. The MGL stage fuses the geometric features from the point cloud with the textural details from the rendered images, producing a unified 3D-aware feature representation. (3) **Language Reasoning and Mask Prediction (grey arrows)**. This stage inputs the 3D-aware feature and text instruction into the structured-output LLM to generate text response and [SEG] token. After that, a mask decoder is utilized to generate the segmentation mask from the [SEG] token.

toward open-vocabulary 3D instance segmentation, while X-RefSeg3D (Qian et al., 2024a) and 3D-STMN (Wu et al., 2024) studied referring 3D segmentation with cross-modal structure and multi-view training. Recent work makes the task more reasoning-oriented: SegPoint (He et al., 2024) unifies instruction-based point-cloud segmentation, Reason3D (Huang et al., 2025b) studies free-form reasoning segmentation, SeqAfford (Yu et al., 2025) decomposes sequential affordance queries, and 2026 works such as MVGGT (Wu et al., 2026a) and 3D-DRES (Chen et al., 2026) improve multi-view grounding and phrase-level segmentation. Our work is related to this progression, but explicitly combines 3D MLLM reasoning with differentiable rendering cues for segmentation.

## 3. Method

### 3.1. Overview

Fig. 3 illustrates the framework (Algorithm 1). Given a point cloud $\mathcal{S}$ and instruction $\mathcal{T}$, the model predicts a segmentation mask $\mathcal{M}$ through three stages:

**Adaptive Viewpoint Position Learning.** First, frozen encoders process $\mathcal{T}$ into $\mathcal{T}_{emb}$ and $\mathcal{S}$ into $\mathbf{F}_p$. These features guide the Adaptive Viewpoint Position Learning Network ($\mathcal{N}_{vp}$) to predict parameters for the Differentiable Rendering module. This module generates query-specific images $\mathcal{M}$, from which an extractor derives multi-view features $\mathbf{F}_v$.

**Multi-modal Group Learning.** As shown by the green arrows, this module fuses geometric features $\mathbf{F}_p$ and tex-

---

**Algorithm 1** The dataflow of TVDRNet

---

**Input:** Point Cloud $\mathcal{S} \in \mathbb{R}^{N \times 6}$; Text Instruction $\mathcal{T}_{ins}$
**Output:** Segmentation Mask $\mathcal{M} \in \{0,1\}^N$; Text Response $\mathcal{T}_{res}$
*Stage 1: Adaptive Viewpoint Position Learning*
Extract features: $\mathcal{T}_{emb} \leftarrow \mathcal{E}_T(\mathcal{T}_{ins})$, $\mathbf{F}_p \leftarrow \mathcal{E}_P(\mathcal{S})$
Predict rendering parameters $\mathbf{u} \leftarrow \mathcal{N}_{vp}(\mathcal{T}_{emb}, \mathbf{F}_p)$ (Eq. 1)
Decompose camera parameters $(u_a, u_e, u_f)$ from $\mathbf{u}$ (Eq. 2)
Render multi-view images $\mathcal{M} \leftarrow \mathcal{R}(u_a, u_e, u_f, \mathcal{S})$ (Eq. 3)
Extract multi-view features $\mathbf{F}_v \leftarrow \mathcal{E}_I(\mathcal{M})$
*Stage 2: Multi-modal Group Learning*
Concatenate: $\mathbf{Z} \leftarrow [\mathbf{F}_p; \mathbf{F}_v] \in \mathbb{R}^{M \times D_p}$
Compute similarity matrix $\mathbf{S} \leftarrow \mathbf{Z}\mathbf{Z}^T$ (Eq. 4)
Compute and normalize coherence scores $\tilde{s}_i$ (Eq. 5, 6)
Assign features to $G$ groups based on score intervals
**for** each group $j = 1$ **to** $G$ **do**
 Compute attention weights and aggregate prototype $g_j$ (Eq. 7, 8)
**end for**
Compute global context $c$ and inter-group attention $\beta_j$
Generate final feature $\mathbf{F}_{3Daware}$ (Eq. 9, 10)
*Stage 3: Language Reasoning and Mask Prediction*
Project features $\mathbf{F}_{3Daware\_proj} \leftarrow \text{Project}(\mathbf{F}_{3Daware})$
Generate response $\mathcal{T}_{res} \leftarrow \text{MLLM}(\mathcal{T}_{emb}, \mathbf{F}_{3Daware\_proj})$
Extract query $h_{seg}$ and compute refined feature $h'_{seg}$
Generate mask $\mathcal{M} \leftarrow \sigma(\text{MLP}(h'_{seg}) \cdot \mathbf{F}_{3Daware}^T)$
**return** $\mathcal{M}, \mathcal{T}_{res}$

---

ture features $\mathbf{F}_v$ to produce unified 3D-aware embeddings $\mathbf{F}_{3Daware}$.

**Language Reasoning and Mask Prediction.** Projected features $\mathbf{F}_{3Daware\_proj}$ and text embeddings $\mathcal{T}_{emb}$ are fed into a LoRA-tuned MLLM. The model generates a response with a [SEG] token. Its hidden state $\mathbf{h}_{seg}$ serves as a query for the Transformer-based Decoder ($\mathcal{M}_{Dec}$), which utilizes $\mathbf{h}_{seg}$ and $\mathbf{F}_{3Daware}$ to generate the final segmentation mask.

### 3.2. Adaptive Viewpoint Position Learning

A primary challenge in 3D reasoning segmentation is spatially interpreting language queries. To this end, TVDRNet introduces a differentiable rendering module in which viewpoint generation is conditioned on the input text description. This design stems from a key insight: the optimal way to view an object depends on what the instruction is asking for. For the same 3D scene, a query like "the top surface" necessitates a high-angle view, whereas "the left leg" requires a low-angle view from the side. The TVDRNet learns this direct mapping from text to camera parameters, ensuring that for any given instruction, it generates a viewpoint specifically tailored to observe the target part with maximum clarity.

The input text is encoded into a feature vector $\mathcal{T}_{emb} \in \mathbb{R}^{1 \times D_t}$, and the point cloud features $\bar{\mathbf{F}}_p \in \mathbb{R}^{N_f \times D_p}$ are processed to compute a global scene feature. These features are then fed into the Adaptive Viewpoint Position Learning

Network (AVPL), $\mathcal{N}_{vp}$, a Multi-Layer Perceptron (MLP) that maps the combined features to the rendering parameter $\mathbf{u}$:

$$\mathbf{u} = \mathcal{N}_{vp}\left(\text{Concat}\left(\text{Norm}(\mathcal{T}_{emb}), \text{Norm}(\text{Mean}(\mathbf{F}_p))\right)\right). \tag{1}$$

After that, the parameter vector $\mathbf{u}$ is then decomposed into rendering camera intrinsic parameters of focal length $u_f$ and rendering camera extrinsic parameters of angles $u_a$ and elevation angles $u_e$:

$$
\begin{aligned}
u_a &= a_{\text{bound}} \cdot \text{sigmoid}\left(\text{Chunk}(\mathbf{u}, 3)[0]\right), \\
u_e &= e_{\text{bound}} \cdot \text{sigmoid}\left(\text{Chunk}(\mathbf{u}, 3)[1]\right), \\
u_f &= f_{\text{bound}} \cdot \text{sigmoid}\left(\text{Chunk}(\mathbf{u}, 3)[2]\right),
\end{aligned}
\tag{2}
$$

where $a_{\text{bound}}$, $e_{\text{bound}}$ and $f_{\text{bound}}$ are hyperparameters defining the maximum boundary of rendering parameters.

With the text-derived camera parameters $(u_a, u_e, u_f)$, a differentiable renderer $\mathcal{R}$ projects the 3D point cloud $\mathcal{S}$ into a set of 2D multi-view images $\mathcal{M}$:

$$\mathcal{M} = \mathcal{R}(u_a, u_e, u_f, \mathcal{S}). \tag{3}$$

The differentiability of 3D rendering allows gradients from the segmentation loss to flow back to the renderers' parameters $(u_a, u_e, u_f)$. This end-to-end process forces the network to spatially interpret the text embedding. For instance, if the views generated for "front hand" lead to a poor segmentation, the gradients will adjust $(u_a, u_e, u_f)$ to map the semantics of "front" to a head-on camera, thus learning to associate texts with an optimal geometric viewpoint.

### 3.3. Multi-modal Group Learning Module

TVDRNet integrates a Multi-modal Group Learning (MGL) module to exploit semantic correlations between point clouds and multi-views. Inspired by (Wu et al., 2026c;b), MGL adaptively groups features $(\mathbf{F}_p, \mathbf{F}_v)$ into $G$ semantic clusters, capturing both shared and modality-specific information. The module processes initial features into an enhanced representation $\mathbf{F}_{3Daware}$ through four steps: similarity calculation, group assignment, intra-group aggregation, and inter-group aggregation.

#### 3.3.1. SIMILARITY MATRIX CALCULATION

We concatenate the extracted point cloud and multi-view features $(\mathbf{F}_p, \mathbf{F}_v)$ into a unified set $\mathbf{Z} = [\mathbf{F}_p; \mathbf{F}_v] \in \mathbb{R}^{M \times D_p}$, where $M = 2N_f$. To measure semantic relationships, we compute a similarity matrix $\mathbf{S} \in \mathbb{R}^{M \times M}$ using L2-normalized features:

$$\mathbf{S} = \mathbf{Z}\mathbf{Z}^T. \tag{4}$$

Here, each element $S_{ij}$ represents the cosine similarity between feature vectors $z_i$ and $z_j$, quantifying their semantic relatedness.

### 3.3.2. SCORING FUNCTION FOR GROUP ASSIGNMENT

To partition features, we first derive a semantic coherence score $s_i$ for each feature $z_i$ by averaging its pairwise similarities from $\mathbf{S}$:

$$s_i = \frac{1}{M-1} \sum_{j \neq i} S_{ij}. \qquad (5)$$

These scores are then normalized to the $[0, 1]$ range via min-max scaling:

$$\tilde{s}_i = \frac{s_i - \min(\mathbf{S})}{\max(\mathbf{S}) - \min(\mathbf{S})}. \qquad (6)$$

We partition $\tilde{s}_i$ into $G$ equal intervals (e.g., $[0, 1/G)$) and assign features to groups based on their scores, ensuring semantic coherence within each cluster.

### 3.3.3. INTRA-GROUP AGGREGATION

For each semantic group $j$, let $\mathbf{Z}_j \in \mathbb{R}^{M_j \times D}$ denote the subset of features. We employ an attention mechanism to identify representative features, computing weights $\alpha_j$ with a learnable query $q_j$:

$$\alpha_{j,i} = \frac{\exp(z_{j,i} \cdot q_j)}{\sum_{k=1}^{M_j} \exp(z_{j,k} \cdot q_j)}, \qquad (7)$$

where $z_{j,i}$ is the $i$-th feature. The group prototype $g_j$ is then aggregated as:

$$g_j = \sum_{i=1}^{M_j} \alpha_{j,i} \cdot z_{j,i}. \qquad (8)$$

This approach dynamically focuses on salient features, yielding discriminative group representations.

### 3.3.4. INTER-GROUP AGGREGATION

To unify the group prototypes $\mathbf{G} \in \mathbb{R}^{G \times D}$ we use hierarchical attention. We first compute a global context $c = \frac{1}{G} \sum_{j=1}^{G} g_j$ and derive attention scores

$$\beta_j = \frac{\exp(g_j \cdot c)}{\sum_{k=1}^{G} \exp(g_k \cdot c)}. \qquad (9)$$

The aggregated representation $\tilde{g} = \sum_{j=1}^{G} \beta_j \cdot g_j$ is transformed and combined with $c$ to yield the final feature:

$$\mathbf{F}_{3Daware} = \mathrm{MLP}(\tilde{g}) + c. \qquad (10)$$

This selectively emphasizes relevant semantic groups to produce the final 3D-aware output.

## 4. Experiment

### 4.1. Experimental Setup and Computation Cost

#### 4.1.1. IMPLEMENTATION DETAILS

The TVDRNet adopts the LLaVA-v1.5-7B (Liu et al., 2024) as MLLM backbone, which utilizes a CLIP ViT-L/14 visual encoder and the Vicuna-7B language encoder. To process the raw 3D data, we employ a pre-trained Sparse 3D U-Net (Sun et al., 2023b) as our point cloud feature extractor. For the language encoder, we employ Low-Rank Adaptation (LoRA) (Hu et al., 2022) for efficient fine-tuning. In AVPL module, which generates multi-views via a differentiable renderer, we set the bounds for azimuth, elevation, and focal length ($a_{\text{bound}}$, $e_{\text{bound}}$, and $f_{\text{bound}}$) to $360°$, $360°$, and $12.0$ respectively. The TVDRNet is trained end-to-end for $100$ epochs using the AdamW optimizer with a cosine annealing scheduler and a learning rate of $2\mathrm{e}{-4}$. All experiments are conducted on 4 NVIDIA L20 GPUs (48GB).

#### 4.1.2. EVALUATION METRICS

According to previous 3D segmentation methods (Feng et al., 2025; Yu et al., 2024b;a), the Mean Intersection over Union (mIoU) and Accuracy at $k$ IoU (Acc@$k$IoU) are utilized to validate the efficiency of TVDRNet on the task of 3D reasoning segmentation and 3D visual grounding. The mIoU metric quantifies the average overlap between the predicted and ground truth 3D volumes. Acc@$k$IoU measures the proportion of predictions where the mask's IoU with the ground truth exceeds a threshold $k$, with $0.25$ and $0.5$ to assess the model's performance at different levels of precision.

### 4.2. Quantitative Comparison

#### 4.2.1. 3D REASONING SEGMENTATION

As presented in Table 1, TVDRNet achieves state-of-the-art performance on both Reason3D and Instruct3D benchmarks. On Reason3D, it attains scores of $55.12$ (Acc@0.25), $42.13$ (Acc@0.50), and $43.92$ (mIoU). Notably, it surpasses the previous best method, OpenMaskDINO3D, by $4.11\%$ in mIoU. On Instruct3D, TVDRNet similarly outperforms the leading MLLM-For3D, achieving gains of $3.0\%$ in Acc@0.50 and $2.7\%$ in mIoU.

Compared to methods operating solely on 3D point clouds and text (rows 3-8), TVDRNet demonstrates substantial improvements, outperforming the Reason3D dataset by $12.72\%$ in mIoU. While prior approaches often struggle with the ambiguity of sparse geometric data, AVPL utilizes text intent to actively guide the differentiable renderer. This process generates semantically aligned visual representations rather than raw geometric features, effectively mitigating the challenges of erroneous spatial ambiguities often

*Table 1.* 3D Reasoning Segmentation Results. The evaluation metric is accuracy at IoU 0.25, IoU 0.5 and mIoU.

| Method | Venue | Input Modality | Reason3D | | | Instruct3D | | |
|---|---|---|---|---|---|---|---|---|
| | | | Acc@0.25 | Acc@0.50 | mIoU | Acc@0.25 | Acc@0.50 | mIoU |
| OpenMaskDINO3D (Zhang, 2025) | Arxiv'25 | RGB+3D+Text | 54.21[#2] | 39.14[#2] | 39.81[#2] | - | - | - |
| MLLM-For3D (Huang et al., 2025a) | Arxiv'25 | RGB+3D+Text | - | - | - | 48.2[#2] | 40.4[#2] | 34.5[#2] |
| Reason3D (Huang et al., 2025b) | 3DV'25 | 3D+Text | 43.21[#3] | 32.10[#3] | 31.20[#3] | 18.35 | 10.55 | 12.43 |
| 3D-STMN (Wu et al., 2024) | AAAI'24 | 3D+Text | 25.43 | 17.78 | 18.23 | - | - | - |
| OpenScene (Peng et al., 2023) | CVPR'23 | 3D+Text | 24.68 | 7.14 | 15.03 | - | - | - |
| OpenMask3D (Takmaz et al., 2023) | NeurIPS'23 | 3D+Text | 20.78 | 6.82 | 13.38 | - | - | - |
| LLM-Grounder (Yang et al., 2024) | ICRA'24 | 3D+Text | - | - | - | 23.7[#3] | 15.6[#3] | 17.2[#3] |
| TGNN (Huang et al., 2021) | AAAI'21 | 3D+Text | - | - | - | 4.76 | 4.76 | 3.51 |
| [†]Avg | - | - | 28.53 | 15.96 | 19.46 | 23.75 | 17.83 | 16.91 |
| TVDRNet (Ours) | - | 3D+Text | **55.12[#1]** | **42.13[#1]** | **43.92[#1]** | **49.5[#1]** | **43.4[#1]** | **37.2[#1]** |

missed by static inputs. By dynamically aligning visual evidence with language intent, TVDRNet facilitates precise grounding, exceeding both specialized and generalist.

Furthermore, TVDRNet surpasses methods that incorporate additional RGB images (rows 1-2), improving Acc@0.50 by 2.99% over OpenMaskDINO3D on Reason3D dataset. Unlike these approaches, which are constrained by fixed and potentially occluded pre-captured views, our method dynamically synthesizes the most informative viewpoints for the specific task. This allows TVDRNet to construct a more holistic 3D perception, achieving superior segmentation accuracy without relying on dense external image.

### 4.2.2. 3D VISUAL GROUNDING

As presented in Table 2, TVDRNet achieves state-of-the-art performance on ScanRefer, attaining 66.2 (Acc@0.25) and 57.9 (Acc@0.50) overall. It surpasses the top task-specific model, UniVLG, and outperforms the leading LMM-based baseline, Reason3D, by a significant 16.0% margin in Acc@0.50. This superiority stems from the AVPL module, which actively optimizes rendering viewpoints to resolve.

### 4.3. Ablation Study

In this section, we conduct ablation experiments to validate the effectiveness of each component in TVDRNet. We focus on three key aspects: (1) the individual contributions of Adaptive Viewpoint Position Learning (AVPL) and Multimodal Group Learning (MGL); (2) the alternative compared with AVPL; (3) the alternative compared with MGL; (4) the alternative combinations of different backbones. All experiments are conducted on the Reason3D dataset for 3D reasoning segmentation.

#### 4.3.1. ANALYSIS EFFICIENCY OF THE DIFFERENT COMPONENTS IN TVDRNET

An ablation study is conducted to analyze the individual contributions of the AVPL and MGL modules. Table 3 presents the results for four distinct model configurations. The baseline model (row 1), which processes only the point cloud and text features with simple concatenation of features to feed into MLLM's text encoder, serves as a performance reference. Configuration (row 2) incorporates the MGL module to fuse point cloud data, yielding a significant performance

*Table 2.* 3D Visual Grounding Results on ScanRefer. The accuracy is evaluated by IoU 0.25 and IoU 0.5.

| Method | Venue | Unique (∼19%) | | Multiple (∼81%) | | Overall | |
|---|---|---|---|---|---|---|---|
| | | Acc@0.25 | Acc@0.50 | Acc@0.25 | Acc@0.50 | Acc@0.25 | Acc@0.50 |
| *Task-specific reasoning* | | | | | | | |
| UniVLG (Jain et al., 2025) | Arxiv'25 | 89.0[#2] | 82.4[#3] | 59.2[#2] | 50.3[#2] | 65.9[#2] | 57.5[#2] |
| Chat-Scene (Huang et al., 2024a) | ECCV'24 | 88.9[#3] | 80.1 | 54.2[#3] | 48.6[#3] | 62.0[#3] | 55.7[#3] |
| AugRefer (Wang et al., 2025) | AAAI'25 | 86.2 | 70.8 | 50.0 | 39.1 | 55.7 | 44.0 |
| X-RefSeg3D (Qian et al., 2024a) | AAAI'24 | - | - | - | - | 40.3 | 33.8 |
| TGNN (Huang et al., 2021) | AAAI'21 | 69.3 | 57.8 | 31.2 | 26.6 | 38.6 | 32.7 |
| ScanRefer (Chen et al., 2020) | ECCV'20 | 67.6 | 44.4 | 31.2 | 20.9 | 38.2 | 25.5 |
| *LMMs-based reasoning* | | | | | | | |
| LLaVA-3D (Zhu et al., 2025a) | Arxiv'25 | - | - | - | - | 50.1 | 42.7 |
| MCLN (Qian et al., 2024b) | ECCV'24 | 84.4 | 68.4 | 49.7 | 38.4 | 54.3 | 42.6 |
| Reason3D (Huang et al., 2025b) | 3DV'25 | 88.4 | 84.2[#2] | 50.5 | 31.7 | 57.9 | 41.9 |
| OpenMaskDINO3D (Zhang, 2025) | Arxiv'25 | - | - | - | - | 42.3 | 28.3 |
| TVDRNet (Ours) | - | **89.2[#1]** | **84.7[#1]** | **60.3[#1]** | **51.2[#1]** | **66.2[#1]** | **57.9[#1]** |

*Table 3.* 3D Reasoning Results on Reason3D. The accuracy is evaluated by IoU 0.25 and IoU 0.5.

| AVPL | MGL | Acc@0.25 | Acc@0.50 | mIoU |
|------|-----|----------|----------|------|
| × | × | 36.15 | 23.68 | 25.35 |
| × | ✓ | 38.31 | 24.12 | 26.47 |
| ✓ | × | 49.26 | 34.24 | 36.57 |
| ✓ | ✓ | **55.12** | **42.13** | **43.92** |

increase and demonstrating the value of multi-modal fusion. To isolate the impact of the adaptive viewpoint mechanism, configuration (row 3) employs AVPL for view generation but replaces MGL with simple feature concatenation. This configuration also outperforms the baseline, underscoring the benefit of generating task-relevant views. The full model (row 4), which integrates both AVPL and MGL, achieves the highest performance, confirming that AVPL's text-guided view selection and MGL's fusion mechanism are complementary for improving segmentation.

### 4.3.2. EFFICIENCY ANALYSIS OF ADAPTIVE VIEWPOINT POSITION LEARNING (AVPL)

The AVPL module enables text-guided adaptive viewpoint generation. We validate its effectiveness through two comparative studies.

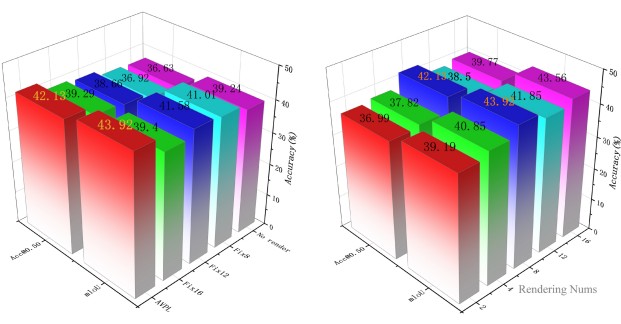

*(a)* Comparing AVPL with alternative rendering configurations.    *(b)* Comparing AVPL with alternative rendering views.

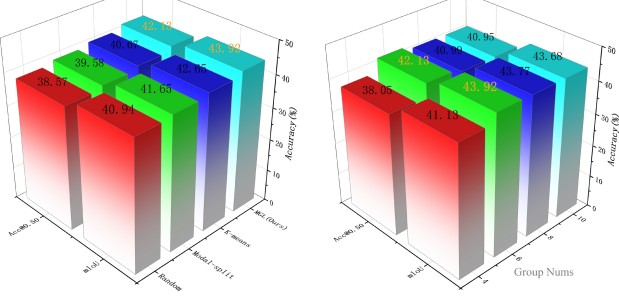

*(c)* Comparing MGL with alternative feature fusion strategy.    *(d)* Comparing MGL with alternative grouping number.

*Figure 4.* Ablation experiments for different design choices of TVDRNet.

**Comparison with Fixed Rendering Configurations.** We benchmark AVPL against static configurations, including No render and fixed viewpoints (8, 12, 16) uniformly distributed. As shown in Fig. 4a, AVPL (using 8 adaptive views) significantly outperforms the best fixed baseline (Fixed-16), achieving 42.13% Acc@0.50 compared to 39.29%. This 2.84% improvement confirms that instruction-aware viewpoint generation provides superior 3D reasoning cues compared to passive.

**Impact of Viewpoint Quantity.** We investigate the optimal number of adaptive viewpoints. Fig. 4b indicates performance peaks at 8 views. Increasing to 16 views degrades Acc@0.50 by 2.36%, due to redundancy-induced overfitting. Conversely, reducing to 2 views drops Acc@0.50 by 5.14%, failing to capture sufficient spatial context. Consequently, 6 views offer the optimal balance between information sufficiency and representational efficiency, adopted for all subsequent.

### 4.3.3. EFFICIENCY ANALYSIS OF MULTI-MODAL GROUP LEARNING (MGL)

The MGL module fuses geometric and visual features. We evaluate its efficiency by analyzing grouping strategies and group.

**Grouping Strategy Comparison.** We compare MGL against Random, Modal-split, and K-means. As shown in Fig. 4c, MGL achieves the best performance, surpassing the second-best method (K-means) by 1.27% in mIoU. This demonstrates that leveraging semantic coherence for grouping enables better cross-modal alignment than rigid geometric.

**Impact of Group Number.** We analyze the effect of the semantic group number $G$. As shown in Fig. 4d, performance peaks at 43.92% mIoU when setting the $G$ as 6. Increasing $G$ further to 8 or 10 yields negligible gains, suggesting that 6 groups provide adequate semantic granularity without introducing.

### 4.3.4. ANALYSIS OF THE COMBINATIONS OF DIFFERENT BACKBONES

We evaluate different backbones for both multimodal feature extraction (from text and images) and point cloud feature extraction to determine the optimal architecture. For the MLLM backbone, we compare LLaVA-1.5-7B (Zhu et al., 2025a) and InternVL2.5-7B (Chen et al., 2024b). For the point cloud feature extractor, we compare Sparse 3D U-Net (Sun et al., 2023a), PointNet++ (Qi et al., 2017), and DGCNN (Wang et al., 2019).

As shown in Fig. 5, the combination of LLaVA-1.5-7B and Sparse 3D U-Net achieves the best performance across all metrics. This is attributed to two key factors: 1) LLaVA-

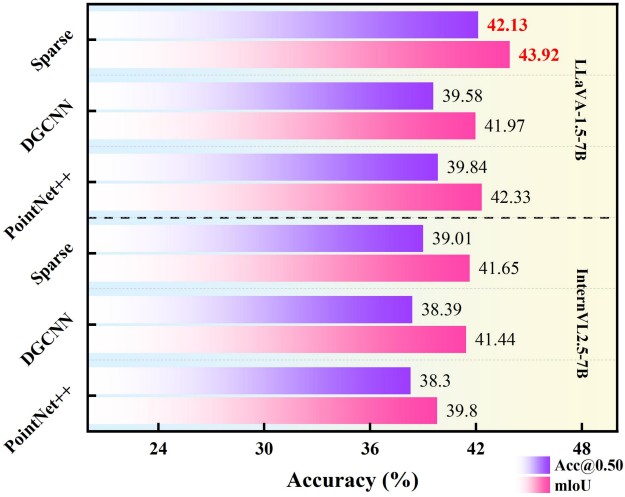

*Figure 5.* The ablation study with the different combinations of backbone.

1.5-7B's architecture and training are highly optimized for visual instruction following, enabling a more precise interpretation of textual commands to guide the viewpoint selection process. 2) Sparse 3D U-Net excels at processing large-scale, sparse 3D scenes, effectively capturing both fine-grained geometric details and broader spatial context, which is crucial for resolving boundary ambiguities. In contrast, PointNet++ and DGCNN, while effective for object-level tasks, tend to lose some scene-level context, resulting in lower performance.

## 5. Conclusion

This paper introduces TVDRNet, an active-vision learning paradigm that addresses key challenges in 3D reasoning segmentation. The TVDRNet enables the network learning 'where to look' based on what the text instruction 'asked to find'. This process generates a semantically aligned visual representation tailored to each query, and then, combined with the original point cloud data, provides a holistic, multimodal input for the reasoning model. The core of the framework features two integral components: the Adaptive Viewpoint Position Learning (AVPL) module, which maps language semantics to optimal camera parameters, and the Multi-modal Group Learning (MGL) module, which fuses geometric and textural features to resolve ambiguities. Extensive experiments on standard benchmarks such as Reason3D, Instruct3D, and ScanRefer demonstrate that TVDRNet achieves competitive performance, effectively mitigating issues of erroneous localization and boundary ambiguity.

## Acknowledgement

We thank the anonymous reviewers for helpful comments on earlier versions of this paper. This research is funded by the National Key R&D Program of China (Grant No.2024YFF0618303), the Scientific and Technological Innovation Project of China Academy of Chinese Medical Sciences(Grant No.CI2023C001YG), and the European Union's Horizon 2024 Research and Innovation Programme for the Marie Skłodowska-Curie Actions (Grant No. 101211118).

## Impact Statement

This work advances multimodal 3D scene understanding by enabling text-driven, adaptive viewpoint optimization for reasoning segmentation. Potential positive impacts include improved accessibility tools for spatial navigation, augmented reality applications, and intelligent interior design systems. However, 3D scene scanning may capture sensitive indoor layouts or personal belongings, raising privacy concerns. Additionally, language-guided models may misinterpret ambiguous instructions or reflect dataset biases, leading to erroneous segmentation in downstream applications. We encourage privacy-preserving rendering techniques and careful human oversight before real-world deployment.

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
