# Supplementary Materials

## 1. Design of TVDRNet's mask generation and training setting

### 1.1. Language Reasoning and Mask Prediction

For vision-language reasoning, the 3D-aware embeddings $\mathbf{F}_{3Daware}$ and the initial text embeddings $\mathcal{T}_{emb}$ are fed into an MLLM's text encoder. The text encoder auto-regressively generates a textual response, which includes a special segmentation token, [SEG]. The hidden state corresponding to this [SEG] token, denoted as $\mathbf{h}_{seg}$, is then extracted. The $\mathbf{h}_{seg}$ encapsulates the high-level semantic instructions for segmentation and serves as the query for the mask decoder.

To generate the final point-wise mask, we utilize a lightweight Transformer-based Mask Decoder, $M_{Dec}$. It takes the language query embedding $\mathbf{h}_{seg}$ and the 3D-aware features $\mathbf{F}_{3Daware}$ as input. Within its cross-attention layers, $\mathbf{h}_{seg}$ functions as the 'Query', while $\mathbf{F}_{3Daware}$ serves as both the 'Key' and 'Value'. This process refines the language query with spatial information from the 3D scene, producing an updated feature $\mathbf{h}'_{seg}$:

$$\mathbf{h}'_{seg} = \text{Attention}(W_Q \mathbf{h}_{seg}, W_K \mathbf{F}_{3Daware}, W_V \mathbf{F}_{3Daware}) \tag{1}$$

where $W_Q, W_K, W_V$ are learnable parameters. The final per-point segmentation mask $M$ is produced by taking the dot product with the 3D-aware features, followed by a sigmoid activation:

$$M = sigmoid\big(\text{ MLP}(h'_{\text{seg}}) \, \mathbf{W} \, \mathbf{F}^T_{3Daware}\big) > 0.5. \tag{2}$$

The $\text{MLP}(h'_{\text{seg}}) \in \mathbb{R}^{1 \times D'}$, is projected by a learnable matrix $\mathbf{W} \in \mathbb{R}^{D' \times D_p}$. This projected feature is then multiplied with the transpose of the per-point cloud features, $\mathbf{F}^T_{3Daware} \in \mathbb{R}^{D_p \times N}$, to yield a score vector. Finally, a binary mask $\mathbf{M} \in \{0, 1\}^N$ is obtained by applying a sigmoid function to this score vector and thresholding the result at 0.5.

### 1.2. Training TVDRNet

The training of TVDRNet is conducted end-to-end. The overall loss function, $L_{TVDRNet}$, comprises two components: the text loss $L_{text}$ for text generation and the segmentation mask loss $L_{mask}$. The overall combination is represented as:

$$L_{TVDRNet} = L_{text} + \lambda L_{mask} \tag{3}$$

where $\lambda$ is a hyperparameter to balance the two task objectives. The text loss, $L_{text}$, embodies the linguistic aspects through an auto-regressive cross-entropy (CE) loss for text generation:

$$L_{text} = \text{CE}(Y_{text}, \hat{Y}_{text}) \tag{4}$$

where $\hat{Y}_{text}$ represents the ground truth word tokens. In addition, the mask loss $L_{mask}$ aims at encouraging the model to generate high-quality segmentation masks. This loss is computed using a combination of a Binary Cross-Entropy (BCE) loss and a Dice loss for all points:

$$L_{mask} = L_{BCE}(M, M_{gt}) + L_{Dice}(M, M_{gt}) \tag{5}$$

where $M$ is the predicted mask and $M_{gt}$ is the ground truth segmentation mask.

During this fine-tuning stage, we keep the main backbones of the feature extractors ($E_T$, $E_P$, $E_I$) and the language model frozen to retain their powerful pre-trained knowledge. The optimization focuses on the trainable parameters, which include the Adaptive Viewpoint Position Learning Network ($N_{vp}$), the Multi-modal Group Learning Module ($M_{MGL}$), the lightweight LoRA modules injected into the language model, and the final Mask Decoder ($M_{Dec}$).

## 2. Further Ablation study and Visualization experiment

### 2.1. The ablations study of Architectural Design Choices and Hyperparameters in TVDRNet

This section investigates the impact of key architectural components and hyperparameters on TVDRNet's overall performance. We focus on two aspects: (1) the choice of pre-trained backbones for vision-language modeling and point cloud encoding, and (2) the balance between text generation and segmentation objectives.

[1]Anonymous Institution, Anonymous City, Anonymous Region, Anonymous Country. Correspondence to: Anonymous Author <anon.email@domain.com>.

Preliminary work. Under review by the International Conference on Machine Learning (ICML). Do not distribute.

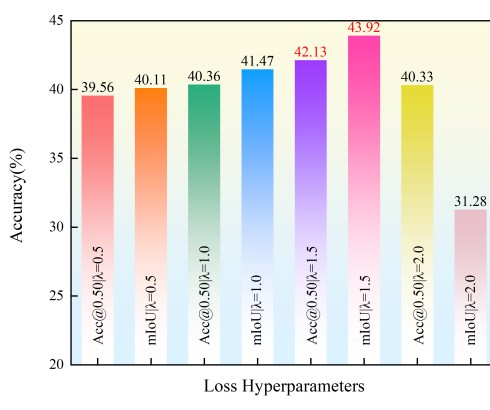

*Figure 1.* The ablation study with the hyperparameter of the loss function.

### 2.1.1. LOSS FUNCTION COMPONENTS

We analyze the sensitivity of the loss-balancing hyperparameter $\lambda$ in Eq. 3. The Figure 1 shows that setting $\lambda = 1.5$ achieves the best trade-off between text generation and segmentation performance. Deviating from this balance—either over-weighting segmentation ($\lambda = 2.0$) or under-weighting it ($\lambda = 0.5$)—leads to noticeable performance drops, especially in Acc@0.50 and mIoU. This indicates that both tasks contribute equally to the final performance and should be treated with balanced importance during training.

### 2.1.2. BACKBONE COMBINATIONS

We evaluate different backbones for both multimodal feature extraction (from text and images) and point cloud feature extraction to determine the optimal architecture. For the MLLM backbone, we compare LLaVA-1.5-7B (Liu et al., 2024) and InternVL2.5-7B (Chen et al., 2024). For the point cloud feature extractor, we compare Sparse 3D U-Net (Sun et al., 2023), PointNet++ (Qi et al., 2017), and DGCNN (Wang et al., 2019).

As shown in Figure 2, the combination of LLaVA-1.5-7B and Sparse 3D U-Net achieves the best performance across all metrics. This is attributed to two key factors: 1) LLaVA-1.5-7B's architecture and training are highly optimized for visual instruction following, enabling a more precise interpretation of textual commands to guide the viewpoint selection process. 2) Sparse 3D U-Net excels at processing large-scale, sparse 3D scenes, effectively capturing both fine-grained geometric details and broader spatial context, which is crucial for resolving boundary ambiguities. In contrast, PointNet++ and DGCNN, while effective for object-level tasks, tend to lose some scene-level context, resulting in lower performance.

## 2.2. Visualization results

The Figure. 3 presents qualitative results on the Reason3D and Instruct3D datasets. These visualizations validate the effectiveness of TVDRNet in addressing key challenges in 3D reasoning segmentation.

The rendered images in Fig. 3(a.2) and Fig. 3(b.2) illustrate the output of the AVPL module. For text instructions such as "water cooler" or "sofa," the AVPL optimizes rendering parameters to generate viewpoints tailored to capture the target object. This instruction-guided viewpoint selection is designed to mitigate erroneous localizations by directing the TVDRNet focus to the relevant region of interest within the 3D scene.

The Grad-CAM (Selvaraju et al., 2017) in Fig. 3(a.3) and Fig. 3(b.3) provide insight into the model's decision-making process. The highlighted areas of high activation correspond closely with the target objects, which suggests that the Multimodal Group Learning (MGL) module effectively fuses geometric information from the point cloud with textural details from the rendered views. This fusion process appears to facilitate the grounding of the language instruction in the visual data.

The segmentation masks in Fig. 3(a.4) and Fig. 3(b.4) exhibit well-defined boundaries, suggesting the model can resolve ambiguity. For instance, the "sofa" is well-delineated from the adjacent wall and floor, which highlights the model's capacity to interpret 3D structures. These qualitative results corroborate the quantitative analysis and suggest the potential of the active-vision paradigm for 3D reasoning segmentation.

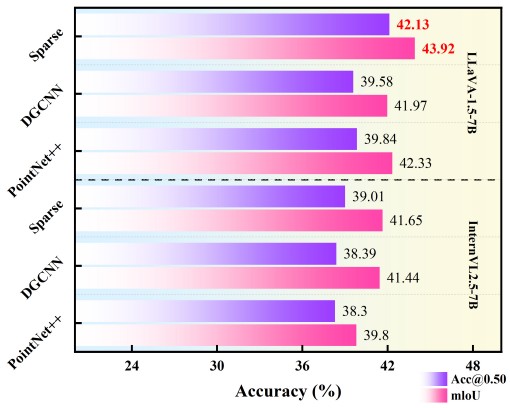

*Figure 2.* The ablation study with the different combinations of backbone.

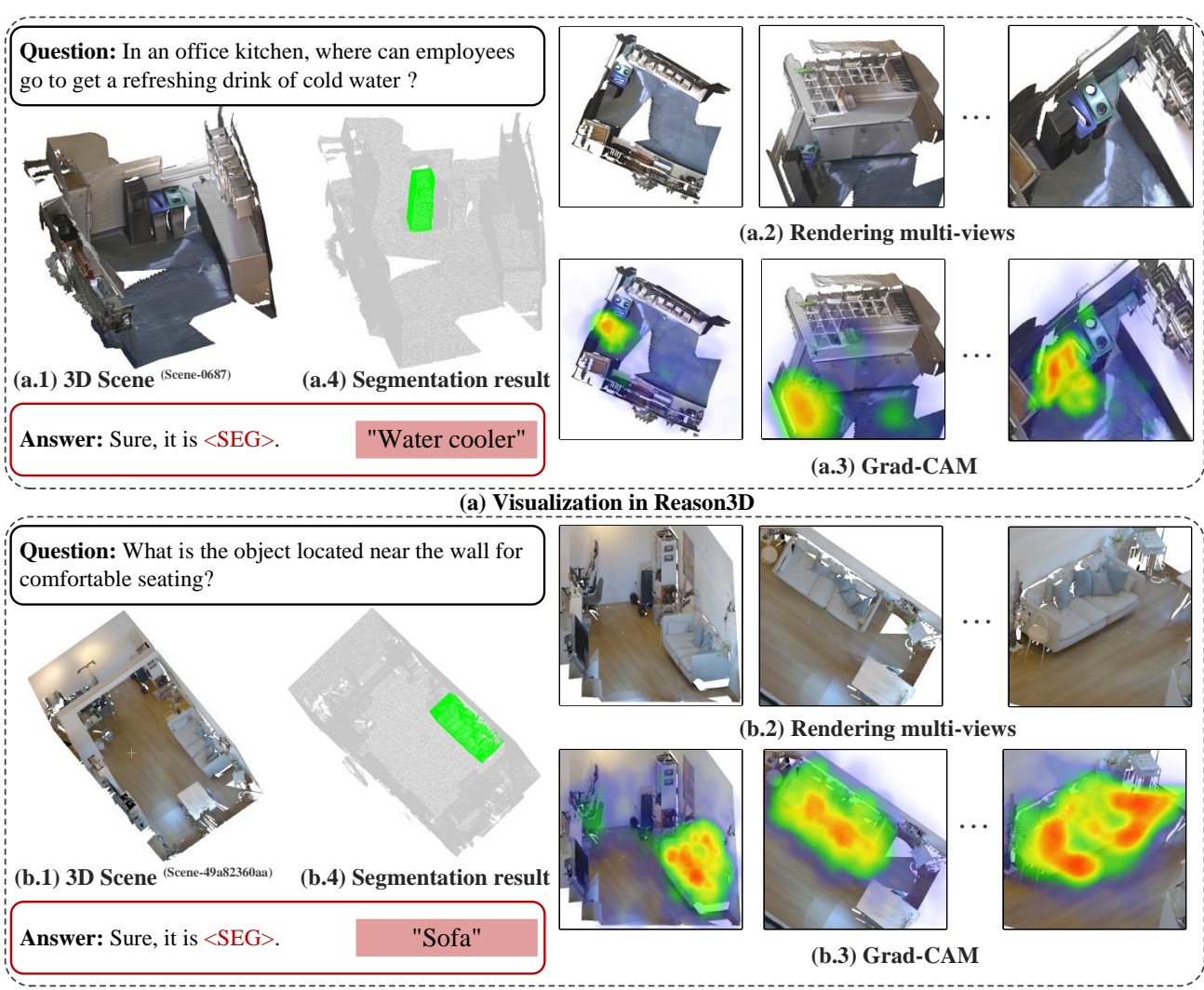

*Figure 3.* Visualization results of TVDRNet on the Reason3D (a) and Instruct3D (b). For each case, (a.1, b.1) are the input 3D point cloud scenes; (a.2, b.2) are the multi-view images generated by the AVPL module, demonstrating its capacity to render task-relevant views guided by the textual instruction; (a.3, b.3) are the Grad-CAM visualizations, which highlight the model's attention, confirming that focus is correctly placed on the objects specified in the text instructions; and (a.4, b.4) are the final segmentation results, characterized by accurate localization and well-defined boundaries.

## 3. Related work

### 3.1. 3D Reasoning Segmentation

The reasoning segmentation task, initially developed for 2D vision by LISA (Lai et al., 2024), has recently been extended to 3D scene understanding. Current research can be broadly categorized into two main streams. The first stream focuses on architectural design to link language reasoning with 3D geometry. These methods (Zhang, 2025; Deng et al., 2025) process point clouds and employ special tokens (e.g., [SEG]) to connect textual outputs with segmentation decoders. Such approaches have been further refined using specialized components, such as hierarchical mask decoders (Huang et al., 2025b), to facilitate coarse-to-fine

object localization. The second stream addresses task definitions and data paradigm expansions. Examples include introducing label-free paradigms via 2D model knowledge transfer (Huang et al., 2025a) and extending the task scope to encompass multi-object scenarios (Jiang et al., 2024).

In contrast to previous efforts that directly process raw 3D data, TVDRNet introduces an intermediate representation step. Rather than requiring the MLLM to interpret unstructured point clouds directly, TVDRNet employs a text-driven differentiable renderer to generate query-specific viewpoints. This process yields a semantically aligned visual representation that serves as input for the MLLM. This learning mechanism aims to mitigate localization and boundary errors that

frequently arise when directly processing unstructured 3D points.

### 3.2. Multimodal Large Language Model in 3D Understanding

The LLaVA (Liu et al., 2023) established a foundational paradigm for extending MLLM capacity in 3D understanding, demonstrating that a pretrained 3D encoder can be effectively coupled with a large language model via a linear projection layer and instruction tuning. This paradigm has been extended to 3D understanding along two primary technical approaches: (1) leveraging 2D MLLM capabilities by lifting features from rendering multi-views into a 3D context through feature aggregation (Hong et al., 2023) or pixel-level alignment (Peng et al., 2023); (2) enabling direct point cloud processing by encoding them into token sequences interpretable by language models (Deng et al., 2025). These architectures have been applied to various 3D tasks, including visual question answering (Azuma et al., 2022; Ma et al., 2022) and language-guided segmentation (Chen et al., 2020; Jiang et al., 2024).

While sharing the goal of language-guided 3D understanding, TVDRNet diverges in its approach to processing 3D scenes. Rather than processing static 3D input, TVDRNet introduces a dynamic, task-driven perception mechanism that actively generates visual representation by employing a text-driven differentiable renderer to produce query-specific viewpoints. This process yields a semantically-aligned and less ambiguous visual input for the MLLM, mitigating challenges such as erroneous localization and boundary ambiguity.

### 3.3. Differentiable Rendering in 3D Understanding

Differentiable rendering (DR) optimizes 3D scene parameters by propagating gradients from training losses (Hamdi et al., 2025). As an Inverse Graphics mechanism (Aristidou et al., 2018), DR infers 3D scene properties from 2D observations, enabling fine-grained optimization of geometry and appearance that enhances the accuracy of task-relevant view synthesis. In the task of 3D shape classification, DR enables backpropagation of gradients from a 2D classifier's loss to optimize rendering parameters for learning task-specific views (Hamdi et al., 2025) or text-guided perspectives for zero-shot learning (Ning et al., 2024). In the task of 3D object pose estimation, DR is utilized through an "analysis-by-synthesis" loop, where the discrepancy between the 3D rendered and 2D images is minimized to refine the pose of the object (Kato et al., 2020; Wang et al., 2021).

While TVDRNet also employs DR for multi-view rendering, it differs from previous efforts by introducing a dynamic, task-driven perception mechanism that does not rely on a static 3D input. TVDRNet generates its visual representation by using a text-driven differentiable renderer to produce query-specific viewpoints. This approach yields a semantically-aligned visual input for the Multi-Modal Large Language Model (MLLM), potentially mitigating challenges such as erroneous localization and boundary ambiguity that can arise when processing raw 3D data.