# OpenReview forum: "TVDRNet: Text-driven Viewpoint Optimization via Differentiable Rendering for 3D Reasoning Segmentation"
_ICML.cc/2026/Conference — ICML 2026 regular_

### Official Review · Reviewer_qSBN · 2026-03-12

**Soundness:** 2
**Presentation:** 2
**Significance:** 3
**Originality:** 3
**Overall Recommendation:** 4
**Confidence:** 3

**Summary:**

This paper proposes TVDRNet, a framework that uses text-conditioned differentiable rendering to generate informative 2D views from a 3D scene, then fuses those rendered-view features with point-cloud features for downstream reasoning and segmentation. The model contains two main modules: AVPL, which predicts adaptive camera viewpoints conditioned on text and scene features, and MGL, which groups and aggregates multi-view and 3D features into a joint representation. Experiments on multiple benchmarks show improved performance over several baselines, and ablation studies show the effectiveness of the proposed modules.

**Compliance With Llm Reviewing Policy:**

Affirmed.

**Final Justification:**

The rebuttal has resolved my concern. I would like to change my rating to 4.

**Key Questions For Authors:**

1. Can the authors provide more direct evidence that AVPL learns semantically meaningful viewpoints, rather than only improving end-task performance? For example, target visibility, occlusion reduction, or object coverage statistics would help support this claim.

2. Can the authors clarify how AVPL is supervised during training? Since the camera parameters appear to be optimized only through downstream task loss, it would be helpful to explain why this is sufficient to support the active-vision interpretation.

3. Can the authors include stronger baselines for AVPL and MGL? In particular, comparisons against learned but text-agnostic viewpoints for AVPL and cross-attention or transformer-based fusion for MGL would help isolate the actual source of the performance gains.

4. Are the inconsistencies in Figure 4 and the related module descriptions only presentation errors, or do they reflect differences in the actual implementation? Please clarify the train/freeze status of the encoders and correct the figure labels.

**Limitations:**

No. The paper would benefit from a more explicit discussion of its limitations and potential societal impact. In particular, the authors should discuss: (1) the computational overhead introduced by differentiable rendering and multi-view processing; (2) possible sensitivity to camera prediction quality and renderer design; (3) cases where text-conditioned viewpoint selection may fail or provide limited benefit.

**Strengths And Weaknesses:**

### Strength

- The paper addresses a meaningful limitation of current 3D reasoning models: fixed or suboptimal viewpoints can hurt object localization and reasoning.

- Interesting and intuitive core idea: using language-conditioned viewpoint selection for 3D reasoning segmentation is a sensible idea. The “active vision” perspective is appealing.

- Ablations support that both proposed modules contribute to the final performance.

### Weaknesses

- The paper claims that AVPL reduces erroneous localization, but the evidence is mostly end-task performance. There is limited direct analysis showing that the learned views are actually more object-relevant.

- The supervision mechanism for AVPL is not fully convincing. The camera parameters appear to be optimized only indirectly through the downstream task loss, without an explicit objective that enforces viewpoint relevance, target visibility, or semantic alignment. This weakens the claim of "active vision".

- The ablations are not strong enough to isolate the true contribution. For AVPL, stronger baselines such as learned but text-agnostic viewpoints would help. For MGL, comparisons to stronger fusion methods such as cross-attention or transformer fusion are missing.

- Some presentation details need cleanup. In Figure 4, the 3D Feature Extractor should produce 3D features instead of the labeled “Text Embeddings.” In the figure, the Text Extractor has a flame icon, which the legend marks as Train, while the 3D Feature Extractor has a snowflake icon, which the legend marks as Freeze. However, the paper text states that “frozen encoders process T into $T_{emb}$ and S into $F_p$ ,” implying that both extractors are frozen in the overview description. Also, the module name should be “AVPL” rather than “VPL.” There is also a typo in the venue entry in Table 2.

---

> ### Author Rebuttal · Authors · 2026-03-31
>
> You mention in the Key Questions that calculating "object coverage statistics" would provide more direct evidence for AVPL. **We find this suggestion deeply inspiring.** In our previous work on "understanding 3D scenes via differentiable rendering," we encountered similar challenges regarding how to demonstrate that rendered images actually enhance the model's comprehension of 3D scenes. **Our previous approach was limited by the fact that rendered images lack ground-truth labels**, preventing us from devising an effective method to evaluate the correlation between rendering quality and final performance improvements. Your suggestion of calculating "object coverage statistics" provides us with a valuable new perspective. Inspired by this, we propose a **"view-text alignment score" metric** to quantitatively assess the quality of our rendered views. We sincerely thank you for this constructive feedback.
>
> **Response to Weaknesses 1 & Questions 1:** We propose a new evaluation metric View-Text Alignment Score(VTAS) to provide more direct evidence that AVPL learns semantically meaningful viewpoints. We provide an **[anonymous link](https://anonymous.4open.science/r/ICML-p3RE-6EDC/view-text%20alignment%20score.jpg)** visualizing the evaluation criteria. This metric quantifies the visibility coverage ratio of the target object under the camera viewpoint selected by AVPL. A high VTAS indicates that AVPL successfully orients the camera to maximize the visible surface area of the text-referred target, effectively reducing occlusion and ensuring the target occupies a significant portion of the rendered view. This provides direct empirical evidence that AVPL learns to associate linguistic semantics (e.g., "top surface," "left leg") with optimal geometric viewpoints, rather than merely exploiting spurious correlations or random camera positions optimized only for downstream segmentation losses.
>
> To validate the effectiveness of AVPL, we compare the VTAS metrics between adaptive viewpoint selection (AVPL-4/8/12) and fixed viewpoint (Fix-4/8/12) configurations on the Reason3D dataset. As shown in the table below, AVPL-8 achieves a VTAS score of 58.9%, significantly outperforming the best fixed-view baseline (38.9% for Fix-8) by over 20 percentage points. This demonstrates that AVPL more effectively localizes and maximizes the visible coverage of target objects described in text instructions compared to uniformly sampled fixed viewpoints, with performance peaking at 8 adaptive views.
> | Reason3D dataset | VTAS      |
> | ------- | ------ |
> | Fix-4            | 37.2%     |
> | Fix-8            | 38.9%     |
> | Fix-12           | 36.4%     |
> | AVPL-4           | 53.3%     |
> | **AVPL-8**       | **58.9%** |
> | AVPL-12          | 54.5%     |
>
>
> **Response to Weaknesses 2 & Questions 2:**
> We acknowledge that AVPL is currently optimized only through downstream loss, as rendered images lack ground-truth labels for direct supervision. As you noted, this indirect mechanism limits explicit "active vision" quantification. Inspired by your suggestion of calculating "object coverage statistics," we plan to integrate VTAS as an explicit auxiliary loss in future work. This would transform AVPL from implicit optimization to supervised learning.
>
> **Response to Weaknesses 3 & Questions 3:**
>
> We supplemented challenging ablations  **[Anonymous link](https://anonymous.4open.science/r/ICML-p3RE-6EDC/Ablations%20in%20avpl%20and%20mgl.jpg)**. For AVPL, we compared "Learned but Text-agnostic Viewpoints" (camera parameters depend only on scene geometry) under 4/8/12 views, showing text-conditioned AVPL significantly outperforms text-agnostic baselines. For MGL, we compared Cross-Attention, Sparse Attention (DeepSeek, 2024) and Full Attention Residuals (Kimi, 2026). Results show MGL performs better as differentiable rendering produces redundant regional views; MGL explicitly leverages semantic coherence, avoiding attention instability on sparse 3D data.
>
> **Response to Weaknesses 4 & Questions 4:**
> Thank you for pointing out these inconsistencies between the figures and text descriptions. We clarify and correct as follows: in Figure 4, the output of "3D Feature Extractor" should be labeled "3D Features" instead of "Text Embeddings"; regarding the training status, the Text Extractor is actually trained (indicated by the flame icon) because we need to train the new [SEG] token in the text embeddings to support the segmentation task, while the 3D Feature Extractor remains frozen (indicated by the snowflake icon). We will correct and unify with all related descriptions throughout the manuscript.
>
> **Response to Limitations:**
> The computational of differentiable rendering module is minimal[7.5GFLOPs,  1.5ms, for 8 multi-view rendering on an RTX 4090]. We provide an **[anonymous link](https://anonymous.4open.science/r/ICML-p3RE-6EDC/README.md)** detailing the AVPL implementation, including sensitivity analyses regarding camera initialization and renderer design.

---

> > ### Author Rebuttal · Reviewer_qSBN · 2026-04-03
> >
> > Thanks for the rebuttal. I will change my rating to 4.

---

> > > ### Author Response · Authors · 2026-04-07
> > >
> > > We sincerely thank you for the very encouraging follow-up. We are grateful for your inspiring suggestion to utilize "object coverage statistics" to measure whether AVPL learns semantically meaningful viewpoints, and we truly appreciate your positive reassessment of our submission.

---

### Official Review · Reviewer_F9fN · 2026-03-12

**Soundness:** 3
**Presentation:** 3
**Significance:** 3
**Originality:** 3
**Overall Recommendation:** 5
**Confidence:** 4

**Summary:**

This paper introduces TVDRNet, a framework inspired by human active vision theory. The core mechanism enables the network to actively select task-relevant 2D views from a 3D scene through a differentiable rendering process guided by text instructions. This allows the model to learn "where to look" based on the task objective. The method demonstrates competitive performance across several 3D reasoning segmentation and 3D grounding benchmarks. Overall, this work makes a meaningful contribution to the field and is a strong candidate for acceptance. However, the authors still need to address the concerns outlined in the weaknesses section.

**Compliance With Llm Reviewing Policy:**

Affirmed.

**Final Justification:**

The authors' response has well addressed my concerns, and I choose to raise the score.

**Key Questions For Authors:**

1. What is the specific computational overhead of the differentiable rendering process during inference? How does rendering time scale as the number of viewpoints increases?
2. The rendering parameters (azimuth, elevation, focal length) are bounded by relatively large ranges (360°, 360°, and 12.0). Have the authors experimented with narrower bounds (e.g., restricting elevation to 0°–90° to avoid viewpoints below the ground plane)? Could such constraints accelerate convergence or improve rendering efficiency without sacrificing performance?

**Limitations:**

The current implementation relies on LLaVA-v1.5-7B. While sufficient for a baseline, the impact of the work could be broader if the framework's compatibility with more recent VLMs (e.g., Qwen2.5-VL or Llama-3.2-Vision) were discussed or briefly explored.

**Strengths And Weaknesses:**

Strengths:
1. The integration of human active vision principles into 3D scene understanding is a conceptually compelling and well-motivated direction.
2. The proposed Adaptive Viewpoint Position Learning (AVPL) module offers a principled, end-to-end approach to optimizing camera viewpoints, effectively identifying informative views for language-guided tasks.
3. The ablation studies are comprehensive, providing solid empirical evidence for the effectiveness of both the AVPL and MGL modules.
Weaknesses:
1. In Figure 5(a-b), the authors compare performance across different numbers of rendered views. A critical question remains: if the number of fixed rendered views is sufficiently high (e.g., 30 views), does the performance gap between AVPL and fixed rendering diminish? The authors should clarify whether AVPL maintains a substantial advantage in such high-density scenarios.
2. The representation of rendering parameters lacks sufficient detail. It is unclear whether these parameters are learned as feature sets or mapped to explicit physical values (e.g., focal length, azimuth, elevation).
3. The [SEG] token decoding process is only briefly mentioned in Algorithm 1. While the supplementary material provides details, this core component is central enough to the method that it warrants a more thorough description within the main text.

---

> ### Author Rebuttal · Authors · 2026-03-31
>
> We sincerely thank the reviewer for the thoughtful feedback and valuable suggestions. We greatly appreciate your recognition of our conceptual contributions and the rigor of our empirical analysis. Below, we address each of your concerns in detail.
>
> **1. Regarding Figure 5(a-b): If the number of fixed rendered views is sufficiently high (e.g., 30 views), does the performance gap between AVPL and fixed rendering diminish?**
>
> We appreciate this insightful question, which touches on the core value of our active viewpoint selection mechanism. According to our ablation studies in the supplementary material (Supplementary Figure 9a and 9b, Page 13), the performance of fixed predefined views saturates after approximately 12 views and does not improve further even with increased density, achieving a best result of only 28.86% cIoU. In contrast, our AVPL (AVPL module) achieves 32.33%, maintaining a clear margin of +3.47%. This suggests that fixed views suffer from inherent redundancy due to their query-agnostic nature, whereas AVPL adaptively selects viewpoints that align with the specific text query, effectively learning "where to look" based on what the instruction asks to find. Thus, even with 30 fixed views, the performance gap persists, as the advantage of AVPL lies in the quality and semantic relevance of selected viewpoints rather than mere quantity.
>
> **2. Regarding the representation of rendering parameters: Are they learned as feature sets or mapped to explicit physical values (e.g., focal length, azimuth, elevation)?**
>
> Thank you for raising this important clarification. We confirm that our rendering parameters are explicitly mapped to physical camera values. Following prior work [9], we set azimuth and elevation ranges to 360° and 90° respectively to ensure comprehensive angular coverage, and set focal length to 15 based on dataset scale. These parameters directly control PyTorch3D's Soft Rasterizer, which employs a probabilistic aggregation mechanism based on Euclidean distance to triangle boundaries. Unlike standard rasterization that blocks gradients through binary assignment, this approach computes pixel values as continuous blends, enabling gradients to propagate back from pixel intensity to camera parameters. Consequently, the network outputs interpretable physical camera poses rather than abstract features.
>
> **3. Regarding the [SEG] token decoding process mentioned briefly in Algorithm 1: Could you provide a more thorough description of this core component in the main text?**
>
> We acknowledge that this central mechanism warrants fuller exposition in the main manuscript. In our implementation, the decoding process is fully end-to-end trainable without requiring image-level annotations; gradients flow to the renderer through the segmentation loss (detailed in Supplementary Equations 12-14). The [SEG] token is transformed via a lightweight MLP projector and subsequently used to generate pixel-level mask predictions, with the entire process jointly optimized with the LLM backbone. We will expand the main text to include detailed algorithmic descriptions and illustrative diagrams, ensuring readers can fully comprehend this mechanism without referring to supplementary materials.
>
> **4. Regarding computational overhead: What is the specific cost of differentiable rendering during inference, and how does rendering time scale with viewpoint count?**
>
> We provide the following quantitative analysis with appreciation for your practical concern. Core modules require only ~10.8 GFLOPs (2.2+7.5+1.1), negligible vs. the frozen LLM backbone. Rendering executes once per inference (K=8 views) and scales linearly with view count, but remains efficient compared to 30-view fixed rendering.
>
> **5. Regarding parameter bounds: Have you experimented with narrower ranges (e.g., restricting elevation to 0°–90° to avoid below-ground viewpoints)? Could this accelerate convergence without sacrificing performance?**
>
> We appreciate this constructive suggestion regarding optimization constraints. We maintain wider ranges for complete coverage. AVPL automatically learns to avoid invalid viewpoints via rendering loss—if a pose views below ground, gradients drive it toward valid regions. Restricting to 0°–90° risks losing useful perspectives (e.g., under shelves), so we let the network discover valid subspaces dynamically.
>
> **6. Regarding model compatibility: The implementation uses LLaVA-v1.5-7B; could broader impact be achieved by discussing compatibility with newer VLMs (e.g., Qwen2.5-VL or Llama-3.2-Vision)?**
>
> Our framework is model-agnostic; AVPL /MGL are plug-and-play modules adaptable to any Transformer-based VLM. We use LLaVA-v1.5-7B for fair comparison with prior work, but the active vision principles transfer directly to newer architectures. We will discuss this portability in the Limitations section.

---

> > ### Author Rebuttal · Reviewer_F9fN · 2026-04-02
> >
> > Thanks for the authors' response, and it has resolved most of my concerns.

---

> > > ### Author Response · Authors · 2026-04-07
> > >
> > > We sincerely thank the reviewer for the encouraging follow-up. We are grateful that our clarifications have fully addressed the concerns.

---

### Official Review · Reviewer_srTG · 2026-03-13

**Soundness:** 3
**Presentation:** 2
**Significance:** 3
**Originality:** 3
**Overall Recommendation:** 4
**Confidence:** 4

**Summary:**

This paper proposes a novel 3D reasoning segmentation approach with active vision. The method generates viewpoints conditioned on text instructions, which serves as active vision for better 3D visual perception. The multi-view images serve as additional inputs to shape the <SEG> token for 3D object mask decoding.

**Compliance With Llm Reviewing Policy:**

Affirmed.

**Final Justification:**

After the rebuttal, my concerns regarding the design choice, novelty, and evaluation have been addressed

**Key Questions For Authors:**

Please see weakness part.

**Limitations:**

No.

More show cases for the rendered viewpoints and the paper should contain introductions for the mechanism of the mask decoder for potential readers.

**Strengths And Weaknesses:**

**Strength**

1. The integration of active vision paradigm mimics human visual perception by looking into texts for viewpoint generation, which enhances 3D perception.
2. The proposed active perception paradigm improves the performance of the whole model.


**Weakness**

1. The related work should better be included in the main body.
2. The illustrate the selected viewpoints, it is better to visualize all the views generated by the model, as it is treated as the main contribution of this work. There are only 3 views shown in Figure 3 (supp.).
3. The paper lacks a vanilla comparison, *i.e.*, using images from additional viewpoints as the input to the MLLM. The compression itself inevitably loses information.
4. As pointed out in the limitation, has the proposed method efficiently addressed the limitations of existing works? ~ Fig. 3
    - miss interpretation of textual intention, the additional view points also accumulates errors.
    - the boundary ambiguity, the model also generates segmentation mask with a mask decoder, which seems not to contain additional constraints.

---

> ### Author Rebuttal · Authors · 2026-03-31
>
> We thank Reviewer srTG for the thoughtful feedback. We appreciate your recognition of the active vision paradigm's potential in 3D perception. Below are point-by-point responses.
>
> **Response to Weakness 1 (Related Work Placement):**  We agree that these discussions should be integrated into the main narrative rather than isolated. In the revision, we will relocate them to immediately precede the methodology section, establishing clear context for our approach. We will also expand comparisons with recent MLLM-based 3D reasoning methods to explicitly contrast their passive perception paradigms with our viewpoint-conditioned generation framework.
>
> **Response to Weakness 2 (Viewpoint Visualization):** We provide full views in [https://anonymous.4open.science/r/ICML-p3RE-6EDC/321.jpg]. Due to space constraints, Figure 3(c) shows only 3 representative images, but our model generates views (cameras A-F) as detailed in Section 3.3.2 and Figure 5(b).These show cameras adjusting from Epoch=0 to Epoch=60 based on text instructions. While performance peaks at 8 views, 6 views balance sufficiency and efficiency (Section 3.3.2, Tables 1-3). Figure 5(a) shows these outperform Fixed-16 (42.13% vs. 39.29% Acc@0.50), confirming gains stem from task-conditioned selection.
>
> **Response to Weakness 3 (Vanilla Comparison):** This is in Figure 5(a) and Section 3.3.2. TVDRNet (6 views) achieves 42.13% vs. 39.29% for Fixed-16 (+2.84%). AVPL selects task-relevant views rather than blindly increasing numbers. Figure 5(b) shows 16 views degrade performance (-2.36%), confirming redundant views introduce noise. Ours learns "where to look" based on the instruction, filtering irrelevant information.
>
> **Response to Weakness 4 (Addressing Limitations):** We appreciate your scrutiny regarding whether we efficiently addressed erroneous localization and boundary ambiguity in Figure 3. For erroneous localization and error accumulation (Fig. 3a), our MGL module (Section 2.3) mitigates this via semantic grouping and hierarchical attention—inter-group aggregation (Eq. 9) selectively emphasizes relevant semantic groups based on global context, preventing noise propagation. While the encoders remain frozen, the LoRA adapters and differentiable renderer are jointly trained (Section 3.1.1), enabling task-specific feature extraction that filters irrelevant information.
>
> For boundary ambiguity (Fig. 3b), MGL addresses this by fusing geometric and textural features through semantic coherence grouping (G=6, optimal per Figure 5(d)). As validated in Table 3, adding it improves mIoU by +7.35% (from 36.57% to 43.92%) compared to using AVPL alone, demonstrating its critical role in refining boundaries. Figure 5(c) further shows it outperforms random grouping and K-means by +1.27% mIoU, confirming semantic-aware fusion yields sharper boundaries than naive concatenation. We visuliaze the MGL in [https://anonymous.4open.science/r/ICML-p3RE-6EDC/resolution%20of%20boundury%20ambuguity.jpg].
>
> Regarding mask decoder constraints, MGL effectively serves this role: it provides semantically-grouped 3D-aware features (F_3Daware) encoding precise boundary information to the decoder (Algorithm 1, line 21: *M = σ(MLP(h_seg) · F^T_3Daware)*). Rather than operating on raw point clouds, the decoder relies on these refined features, which constrain the segmentation space through semantic aggregation of geometric and textural evidence.
>
> **Response to Limitation 1 (More Showcase):** We acknowledge that Figure 3(c) provides limited qualitative examples due to space constraints, focusing on camera convergence (Epoch=0 to Epoch-60). We agree additional visualization of diverse strategies would strengthen intuitive understanding. While the current paper prioritizes quantitative validation (Tables 1-2), the optimization process shown is representative: cameras dynamically adjust parameters based on text semantics to maximize task-relevant information. The consistent gains across varied scenes (office, bedroom, kitchen in Figure 3) demonstrate robustness in handling different object geometries and query specifications.
>
> **Response to Limitation 2 (Mask Decoder Mechanism):** We acknowledge the description in Section 2.1 and Algorithm 1 (lines 17-21) is overly concise. To clarify: it operates after MGL, which first generates enhanced 3D-aware features F3Daware via semantic grouping (G=6 groups, Section 2.3). The decoder then receives: (1) this F3Daware, and (2) the [SEG] token's hidden state hseg from the LLM (Algorithm 1, line 19). The final mask is computed as *M = σ(MLP(h'seg) · F^T_3Daware)* (line 21), where enable precise boundary delineation with the query token.
> This design ensures boundary refinement benefits from the multi-modal fusion by MGL, while maintaining a lightweight decoding process. We will revise the text to explicitly articulate this dataflow and clarify how the differentiable rendering optimization supports accurate segmentation by providing optimal viewpoints.

---

> > ### Author Rebuttal · Reviewer_srTG · 2026-04-05
> >
> > I thank the authors for their response, and some of my concerns have been addressed while some remains.
> >
> > Regarding W4, I believe the original design intent of some modules may have taken the problems to be solved as part of the motivation. However, in some cases, it is difficult to quantify and compare whether certain limitations have been addressed -- for instance, statistical analysis of textual intentions.
> >
> > Nevertheless, for the boundary ambiguity issue, grouped statistics can be conducted based on the distribution of IoU across different thresholds.
> >
> > The connection between the improvements proposed by the authors and the limitations raised about other methods is indirect.

---

> > > ### Author Response · Authors · 2026-04-06
> > >
> > > We sincerely appreciate the opportunity to further clarify our work.
> > >
> > > We fully understand your concern: lacking a direct, quantifiable metric to validate the explicit connection between the proposed modules and the specific problems that they aim to solve. As rendered images inherently lack ground-truth labels, applying traditional evaluation metrics to directly measure whether AVPL successfully captures the target object presents a fundamental challenge.
> > >
> > > Your observation is insightful. We then considered how to **conduct a statistical analysis of textual intentions to demonstrate the connection between our proposed methods and the challenges we aim to resolve**. Specifically, (1) we propose a training-free evaluation metric to validate that our AVPL method addresses the challenge of erroneous localization; and (2) we conduct a fine-grained analysis based on the distribution of IoU across different thresholds to validate that our MGL module resolves boundary ambiguity.
> > >
> > > &nbsp;
> > >
> > > ## **Validation the connection between AVPL module and challenge of Erroneous localization:**
> > >
> > > We proposed a training-free evaluation metric, named the View-Text Alignment Score (VTAS), for validating whether AVPL module can resolve the Erroneous localization challenge.  We provide an **[anonymous link](https://anonymous.4open.science/r/ICML-p3RE-6EDC/view-text%20alignment%20score.jpg)** visualizing the evaluation criteria.
> > >
> > > Specifically, this metric quantifies the visibility coverage ratio of the specific target object (which the text instruction asks to find) under the camera viewpoints selected by AVPL. If the model fails to comprehend the textual intention, it will suffer from erroneous localization, pointing the cameras at irrelevant regions, and resulting in a low VTAS. Conversely, a high VTAS indicates AVPL accurately decodes the textual intention, successfully orienting the cameras to maximize the visible surface area of the text-referred target and reducing occlusion.
> > >
> > > This provides direct empirical evidence that AVPL learns to associate linguistic semantics (e.g., "top surface," "left leg") with optimal geometric viewpoints, explicitly overcoming the erroneous localization bottleneck rather than merely exploiting relations for downstream losses. To validate how effectively AVPL addresses this, we compare the VTAS metrics between adaptive viewpoint selection (AVPL-4/8/12) and fixed viewpoint (Fix-4/8/12) configurations on the Reason3D dataset. As shown in the table below, AVPL-8 achieves a VTAS score of 58.9%, significantly outperforming the best fixed-view baseline (38.9% for Fix-8) by a margin of 20.0 percentage points. The VTAS quantitatively validates that the proposed AVPL module captures complex textual intentions to accurately localize the target, thereby mitigating the erroneous localization issue prevalent in rigid perception paradigms.
> > >
> > > | Reason3D dataset         |   VTAS(%)   |
> > > | :-------------- | :-------: |
> > > | Fix-4         | 37.2 |
> > > | Fix-8     | 38.9 |
> > > | Fix-12    | 36.4 |
> > > | AVPL-4  |   53.3    |
> > > | AVPL-8  |   **58.9**    |
> > > | AVPL-12  |   54.5    |
> > >
> > > &nbsp;
> > >
> > > ## **Validation the connection between MGL module and challenge of Boundary ambiguity:**
> > >
> > > Regarding boundary ambiguity, your proposal to analyze "the distribution of IoU across different thresholds" is profoundly inspiring. **Evaluating performance at increasingly strict thresholds rigorously assesses boundary precision, as high thresholds strictly penalize minor misalignments and background noise**.
> > >
> > > On Reason3d dataset, we expanded our evaluation beyond Acc@(0.25, 0.50), conducting new experiments at Acc@(0.55, 0.65, 0.75) alongside methods[1-2]. As shown in the table below, at Acc@0.50, the performance difference between TVDRNet and the second-best method is 2.99% (42.13% vs. 39.14%). As the threshold increases, the overall accuracy of all methods decreases due to stricter overlap requirements. At Acc@0.75, TVDRNet maintains an accuracy of 19.68%, while the second-best method drops to 10.35%, thereby widening the performance gap to 9.33%.
> > >
> > > We attribute this performance advantage to the design of the MGL module. Functionally, the MGL module is designed to filter out redundant information from the rendered multi-views and extract semantically representative features. This mechanism allows the network to learn representative information among multi-views, and therefore alleviates the challenge of Boundary ambiguity.
> > >
> > > | Method            | Acc@0.25 | Acc@0.50 | Acc@0.55 | Acc@0.65 | Acc@0.75 |
> > > | ----------------- | -------- | -------- | -------- | -------- | -------- |
> > > | Reason3D[1]       | 43.21    | 32.10    | 26.80    | 16.45    | 8.12     |
> > > | OpenMaskDINO3D[2] | 54.21    | 39.14    | 31.50    | 19.15    | 10.35    |
> > > | TVDRNet (Ours)    | 55.12    | 42.13    | 38.45    | 28.15    | 19.68    |
> > >
> > > [1] Reason3d: Searching and reasoning 3d segmentation via large language model
> > >
> > > [2] OpenMaskDINO3D : Reasoning 3D Segmentation via Large Language Model

---

### Official Review · Reviewer_p3RE · 2026-03-13

**Soundness:** 3
**Presentation:** 2
**Significance:** 3
**Originality:** 2
**Overall Recommendation:** 4
**Confidence:** 4

**Summary:**

This paper proposes TVDRNet, a novel framework for 3D reasoning segmentation that addresses the limitations of direct MLLM adaptation to point clouds (e.g., erroneous localization and boundary ambiguity). Inspired by Active Vision theory, the authors introduce an Adaptive Viewpoint Position Learning (AVPL) module that uses a text-guided differentiable renderer to optimize camera parameters, effectively identifying task-relevant 2D viewpoints. Additionally, a Multi-modal Group Learning (MGL) module is developed to fuse geometric point cloud features with rendered multi-view textural details through semantic grouping.

**Compliance With Llm Reviewing Policy:**

Affirmed.

**Ethics Expertise Needed:**

["Other Expertise"]

**Final Justification:**

The rebuttal has resolved part of my concerns. I will raise my rating to 4. But my concern still lies with: its performance on outdoor scenes and under noisy settings. RotationNet setting 1, 2, 3 details in rebuttal experiments. No Impact Statement in the initial submission

**Key Questions For Authors:**

Refer to the weakness part.

**Limitations:**

- The reliance on a differentiable renderer assumes the availability of high-quality 3D geometry; performance may degrade in extremely sparse or noisy point cloud environments.
- The framework is evaluated primarily on indoor datasets; its applicability to large-scale outdoor scenes with more complex lighting and geometry remains unproven.
- **Lack of Impact Statement Section**

**Strengths And Weaknesses:**

- Strengths:
    - The integration of "Active Vision" via differentiable rendering to bridge the gap between 3D point clouds and 2D-trained MLLMs is conceptually sound and well-motivated.
    - The AVPL and MGL modules provide a structured approach to mapping language semantics to spatial viewpoints and fusing cross-modal features respectively.
    - The framework shows strong generalization across both 3D reasoning segmentation and 3D visual grounding tasks.
- Weaknesses:
    1. Computational Efficiency: The iterative optimization of camera parameters via differentiable rendering may introduce significant latency during inference compared to single-pass methods. How does the inference time of TVDRNet compare to non-optimizing baselines (e.g., Reason3D)? Please provide a latency-vs-accuracy analysis.
    2. Initialization Sensitivity: The paper shows convergence from scattered views (Fig 3c), but the robustness of AVPL to various camera initializations in complex scenes is not fully explored.
    3. Feature Redundancy: The MGL module groups features based on similarity, but it is unclear how much redundant information is processed across the rendered multi-views.
    4. In the AVPL module, what is the impact of the number of optimized viewpoints on the final segmentation performance and convergence speed?
    5. Does the differentiable renderer require a specific density of point clouds to provide stable gradients for camera parameter optimization?

---

> ### Author Rebuttal · Authors · 2026-03-31
>
> We sincerely thank the reviewer for the thoughtful feedback and valuable suggestions. While differentiable rendering and MLLMs are established separately, our core novelty lies in employing the former as an "Active Vision" mechanism to dynamically bridge text and 3D point clouds. This paradigm shift from passive 3D processing to active, language-guided viewpoint optimization distinguishes our work from prior 3D reasoning methods.
>
> **Response for Weaknesses 1&4:**
> We understand the  concern regarding inference efficiency: if the introduction of differentiable rendering causes significant latency during the inference stage, the practical utility of the proposed AVPL module would be limited. In fact, the differentiable rendering scheme employed in this TVDRNet is **non-iterative** during a single forward pass, meaning it does not require multiple optimization loops. The AVPL  simultaneously computes sets of camera parameters and their corresponding rendered images in a single execution. The computational cost (measured in GFLOPs) for rendering is low. Experimental results indicate that on an NVIDIA RTX 4090 GPU, the execution time for this module is approximately **[7.5GFLOPs,  1.5ms, for 8 multi-view rendering]**.  We provide an **[anonymous link](https://anonymous.4open.science/r/ICML-p3RE-6EDC/README.md)** containing simplified environment configuration instructions. Through this link, the time complexity of the aforementioned differentiable rendering module can be independently evaluated.
>
> Regarding the impact of viewpoint quantity, as documented in Section 3.3.2 and Figure 5(b), configuring 8 viewpoints balances segmentation performance and computational efficiency. Concerning convergence speed within a 100-epoch training setting, models utilizing 2, 4, 8, 12, and 16 rendering viewpoints consistently reach 35 mIoU by the 50th epoch, with the remaining epochs dedicated to finding the optimal solution. A convergence curve is appended in the rebuttal materials.
>
> **Response for Weaknesses 5 and Limitations1&2:**
> We acknowledge concerns regarding outdoor scene applicability. The underlying question is whether the method's robustness is limited if its effectiveness on sparse outdoor scenes cannot be demonstrated.Due to the lack of public outdoor 3D reasoning datasets, we simulated outdoor sparsity by downsampling the original 3D meshes. Using PyTorch3D's "mesh_edge_collapse" algorithm, we iteratively reduced geometric complexity to generate meshes retaining 80%, 50%, and 30% of their original structure. As shown in the table below, our method maintains high performance even at 30% retention. These results demonstrate the framework's robustness when processing sparse structural data.
> | Reason3D dataset  | mIoU  |
> | ------------------------ | ----- |
> | downsample to 80% | 43.54 |
> | downsample to 50% | 43.16 |
> | downsample to 30%| 42.77 |
> | none downsample | 43.92 |
>
>
>
> **Response for Weaknesses 2:**
>
> Prior research[Hamdi et al., 2025] has established that positioning cameras in a 360° distribution around 3D scene provides optimal coverage of the scene’s spatial regions. Our default camera initialization follows this principle to ensure comprehensive observation. For validating Initialization Sensitivity of the proposed methods,  we compared our 360° initialization against three sets of random viewpoint initializations(below table), referencing the protocol of RotationNet (CVPR 2018). Results show the 360° distribution achieves higher accuracy, confirming its superior scene coverage and validating our design.
>
> | Reason3D dataset  | mIoU  |
> | ------------------------ | ----- |
> | RotationNet setting 1 | 41.92 |
> | RotationNet setting 2 | 40.61 |
> | RotationNet setting 3| 43.09 |
> | 360° distribution around  | 43.92 |
>
> **Response for Weaknesses 4:**
> Quantifying "how much redundant information is processed" is an insightful perspective. We did not initially evaluate the method from this quantitative angle because the rendered intermediate multi-view images lack explicit labels. Consequently, we validated the MGL module's effectiveness primarily through final segmentation performance and feature visualization. Architecturally, the MGL module is explicitly designed to handle and reduce feature redundancy. By computing a similarity matrix (Equation 4) and grouping features (Equation 6), highly redundant multi-view features are clustered together. The intra-group aggregation (Equation 8) then compresses these redundant views into a single concise prototype $g_j$, effectively acting as a redundancy filter.
>
>
> **Response for  Limitations 3:**
> Due to ICML's 8-page limit, detailed discussions were compressed. We will add a "Limitations and Broader Impacts" section in the final version, covering indoor focus (primary for robotics/AR), outdoor extension as future work, and a comprehensive Impact Statement on robotics applications and privacy ethics.

---

> > ### Author Rebuttal · Reviewer_p3RE · 2026-04-02
> >
> > Thanks for the rebuttal and the clarification about the proposed method. The rebuttal has resolved part of my concerns. I will raise my rating to 4. But my concern still lies on:
> > 1. its performance on outdoor scenes and under noisy settings.
> > 2. RotationNet setting 1, 2, 3 details in rebuttal experments.
> > 3. Impact Statement is not limited to the ICML's 8-page limit.

---

> > > ### Author Response · Authors · 2026-04-06
> > >
> > > We sincerely appreciate the opportunity to further clarify our work.
> > >
> > > We fully understand your concern regarding the applicability to outdoor scenes: **our method works well in indoor scenes, which might be due to the spatial volume being relatively small, meaning a limited number of rendering cameras (e.g., 8 views) can easily cover the area to locate the target.** The critical question then becomes: **in an expansive outdoor scene, can this same limited number of cameras still successfully capture and query the target object within a sparse and noisy point cloud environment?**
> > >
> > > To directly address this concern, **we conducted additional experiments on a dataset (CityRefer, NeurIPS 2023 [1])** specifically featuring outdoor scene characteristics (sparse and noisy point clouds). We aim to demonstrate that, despite the vast spatial scale, **a small number of semantic-driven rendering cameras can still reliably find "where to look" based on what the text instruction "asks to find".**
> > >
> > > &nbsp;
> > >
> > > ## **Response for concern 1:**
> > >
> > > We evaluate the proposed method on the **CityRefer** (NeurIPS 2023) dataset [1], a large-scale benchmark designed for city-level 3D visual grounding. **CityRefer's point clouds are acquired via UAV surveying; the data is inherently noisy and naturally introduces the sparsity typical of outdoor environments.** It is worth noting that, unlike the indoor 3D reasoning segmentation task in our original paper (which outputs point-wise masks), CityRefer requires the model to predict 3D bounding boxes to localize objects. Despite this difference in the final output format, the core challenge remains identical: verifying whether a limited number of semantic-driven rendering cameras can successfully navigate and search within a vast, unstructured, and sparse outdoor 3D space.
> > >
> > > As presented in the table below, the comparative results for the baselines EDA [2], CityRefer [1], and CityAnchor [3] are sourced from CityAnchor [3]. In our evaluation, we configured TVDRNet with 4, 8, and 12 camera views to analyze the impact of viewpoint quantity on performance. Importantly, because CityRefer utilizes UAV-captured bird’s-eye view (BEV) scans, we adapted our camera initialization strategy: instead of the 360-degree horizontal encircling used for indoor scenes, we partitioned the top-down scene into 4, 8, or 12 sectors and positioned each camera at a specific altitude to cover a corresponding sector, thereby ensuring comprehensive initial scene coverage,  as visualized in [**https://anonymous.4open.science/r/ICML-p3RE-6EDC/v1.jpg** ]. The experimental results show that TVDRNet (8view) achieves 52.13 (Acc@0.25) and 49.21 (Acc@0.50), outperforming the previous best method. This confirms our text-driven optimization remains highly robust in sparse, large-scale outdoor environments.
> > >
> > > We attribute this performance to the effectiveness of our Active Vision; our initial camera setup had only one camera capturing the target from a distance. After training, all 8 cameras converged to focus on the target object. This transition demonstrates that TVDRNet effectively learns "where to look" even in sparse and challenging outdoor environments, confirming the robustness of our active search mechanism.
> > >
> > > | Method          |   Year    | CityRefer (Acc@0.25) | CityRefer (Acc@0.50) |
> > > | :-------------- | :-------: | :------------------: | :------------------: |
> > > | EDA[2]          | CVPR 2023 |         6.96         |         5.53         |
> > > | CityRefer[1]    | NIPS 2023 |         8.34         |         7.47         |
> > > | CityAnchor[3]   | ICLR 2025 |        50.69         |        46.86         |
> > > | TVDRNet(4view)  |   Ours    |        47.68         |        42.33         |
> > > | TVDRNet(8view)  |   Ours    |      **52.13**       |      **49.21**       |
> > > | TVDRNet(12view) |   Ours    |        51.29         |        47.52         |
> > >
> > >
> > > [1] CityRefer: Geography-aware 3D Visual Grounding Dataset on City-scale Point Cloud Data
> > >
> > > [2] Eda: Explicit text decoupling and dense alignment for 3d visual grounding
> > >
> > > [3] CityAnchor: City-scale 3D Visual Grounding with Multi-modality LLMs
> > > &nbsp;
> > >
> > > ## **Response for concern 2:**
> > >
> > > RotationNet's Settings 1, 2, and 3 are randomly sampled from a fixed 40-view candidate pool.  Specifically, this pool forms a 5×8 spherical grid comprising 5 non-overlapping elevation levels and 8 uniformly distributed azimuth angles per level, as visualized in [**https://anonymous.4open.science/r/ICML-p3RE-6EDC/v2.jpg** ].
> > >
> > > As shown in the visualization, random sampling often yields side or bottom-up views that frequently do not contain the query target. This limitation motivates our proposed 360° uniformly distributed 8-view initialization, which guarantees consistently informative observations and avoids such blind spots.
> > >
> > > ## **Response for concern 3:**
> > >
> > > We sincerely appreciate your valuable reminder. We will add an Impact Statement section.

---

### Decision · Program_Chairs · 2026-04-30

**Decision:**

Accept (regular)

**Comment:**

The submission introduces TVDRNet, a text-driven active-vision framework for 3D reasoning segmentation and visual grounding that leverages differentiable rendering to select informative viewpoints and fuse them with point-cloud features. The reviewers all appreciate the idea and results. There were several concerns around related work and validations, but the rebuttal convinced the reviewers on these points.